# Nutrition, Physical Activity and Smoking Habit in the Italian General Adult Population: CUORE Project Health Examination Survey 2018–2019

**DOI:** 10.3390/healthcare12040475

**Published:** 2024-02-15

**Authors:** Chiara Donfrancesco, Brigitta Buttari, Benedetta Marcozzi, Sabina Sieri, Anna Di Lonardo, Cinzia Lo Noce, Elisabetta Profumo, Francesca Vespasiano, Claudia Agnoli, Serena Vannucchi, Marco Silano, Daniela Galeone, Paolo Bellisario, Francesco Vaia, Luigi Palmieri

**Affiliations:** 1Istituto Superiore di Sanità, 00161 Rome, Italy; brigitta.buttari@iss.it (B.B.); benedetta.marcozzi@iss.it (B.M.); anna.dilonardo@iss.it (A.D.L.); cinzia.lonoce@iss.it (C.L.N.); elisabetta.profumo@iss.it (E.P.); francesca.vespasiano@iss.it (F.V.); serena.vannucchi@iss.it (S.V.); marco.silano@iss.it (M.S.); luigi.palmieri@iss.it (L.P.); 2Fondazione IRCCS, Istituto Nazionale dei Tumori, 20133 Milan, Italy; sabina.sieri@istitutotumori.mi.it (S.S.); claudia.agnoli@istitutotumori.mi.it (C.A.); 3Italian Ministry of Health, 00144 Rome, Italy; d.galeone@sanita.it (D.G.); p.bellisario@sanita.it (P.B.);

**Keywords:** lifestyle, smoking habit, physical activity, nutrition, epidemiology, non-communicable diseases

## Abstract

Background: Tobacco consumption, incorrect nutrition and insufficient physical activity/sedentariness represent modifiable NCDs risk factors in Western countries. To evaluate recent lifestyle indicators in Italy, data from the national Health Examination Survey (HES), implemented in 2018–2019 within the CUORE Project, were assessed. Methods: Age–sex standardized results from random samples of Italian general population (35–74 years) were reported by sex, age-class, educational level and geographical area. From 2106 participants, 2090 were considered for smoking habit, 2016 for physical activity and 1578 for nutrition. Standardized questionnaires were used for smoking habit and physical activity, and the EPIC questionnaire for nutrition. Results: Total cigarette current smokers were 23% in men and 19% in women; sedentariness during leisure time was 34% in men and 45% in women and at work 45% and 47% in men and women, respectively. Prevalence of balanced eating behaviours for vegetables was 28% in men and 39% in women; and for fruits 50% and 52%, respectively; prevalence of correct lifestyle (not smoker, regular physical activity and following at least five correct eating behaviours) was 7% and 12% for men and women, respectively. Conclusions: In 2018–2019, levels of unhealthy lifestyles were found to be still epidemic and basically stable compared to 10 years earlier (slight smoking habit decrease, slight sedentariness increase and slight nutrition improvements); intersectoral strategies and monitoring need to be continued.

## 1. Introduction

Non-communicable diseases (NCDs), such as cardiovascular diseases, cancers, chronic respiratory diseases, diabetes and musculoskeletal disorders, remain the leading causes of death worldwide [1]. Tobacco consumption, incorrect eating habits, insufficient physical activity/sedentariness and risky and harmful alcohol consumption, together with the characteristics of the environment and the social, economic and cultural context represent the main modifiable risk factors in Western countries [1].

These main behavioural risk factors act directly and indirectly to increase the risk of NCDs since they can lead to intermediate risk factors such as arterial hypertension, overweight/obesity, dyslipidaemia, hyperglycaemia and precancerous and early cancerous precancerous lesions. To combat global mortality from NCDs, at the Sixty-sixth World Health Assembly in 2013, Member States developed a Global Plan of Action for 2013–2020, extended to 2030, setting global targets; they included the achievement of nine global targets regarding overall mortality from NCDs, harmful use of alcohol, insufficient physical activity/sedentariness, intake of salt/sodium, current tobacco use, raised blood pressure, diabetes and obesity, heart attacks and strokes drug therapy and counselling for prevention and basic technologies and essential medicines to treat major NCDs [1,2,3].

In Italy, in 2017, about one-third of all deaths could be attributed to behavioural risk factors, such as unhealthy nutrition (16% of death), tobacco smoking (14%), alcohol consumption (4%) and physical inactivity (3%) [4].

To intervene on the four main modifiable risk factors of NCDs (tobacco consumption, sedentary lifestyle/low physical activity, risky alcohol consumption and poor diet), in Italy, the program “Gaining Health: making healthy choices easy” was implemented together with the National Preventive Plans (NPPs) through actions and policies adopting an intersectoral vision. The WHO recommended improving country-level surveillance and monitoring as a priority in the fight against NCDs, and also providing data disaggregated by age, gender and socioeconomic groups [1,2]. Monitoring should provide internationally comparable assessments of the trends in NCDs and related risk factors over time, help to benchmark the situation in individual countries versus others in the same region or development category and provide a support for advocacy, policy development and coordinated action [1,2].

In Italy, national Health Examination Surveys (HESs) have been conducted since 1998. The assessment of tobacco use, physical inactivity and dietary habits was carried out in 1998–2002 and 2008–2012 in the Italian adult general population through national HESs implemented within the CUORE Project [5].

In order to provide more recent lifestyles indicators, data collected in 2018–2019 in random samples of Italian adult general population, through the national HES implemented within the CUORE Project, were analysed and reported by sex, age-classes, educational level and geographical area.

## 2. Materials and Methods

### 2.1. Sampling

From April 2018 to December 2019, the Italian National Institute of Health (Istituto Superiore di Sanità-ISS), within the CUORE Project, conducted a HES in 10 Regions (out of 20) chosen in the North, Centre and South of Italy, enrolling a sample of 100 men and 100 women aged 35–74 years in each examined region (participation rate 40%) randomly selected from the resident registries and stratified by sex and age-group (35–44, 45–54, 55–64 and 65–74 years) [6,7,8,9]. The HES was approved by the Ethical Committee of the ISS; all participants received an invitation letter and an informative note by ordinary postal service and signed an informed consent at the time of the visit. The HES is recognized within the Italian National Statistical Program and within the European HES collaboration [10,11]. The HES 2018–2019—CUORE Project used international standardized procedures and methods for measurements and data collection [6,7,8,9].

### 2.2. Smoking Habit Data Collection

The smoking questionnaire assessed current and former smokers. Current smoker was defined as a person who smokes one or more cigarettes per day. A former smoker is defined as a person who has quit smoking for at least 12 months. Number of cigarettes consumption was recorded for current smokers, as well as type of cigarettes (packaged, handmade or electronic).

### 2.3. Physical Activity Data Collection

Data on physical activity were collected through a questionnaire including four levels of exercise (sedentariness, mild, moderate and heavy), separately for work and leisure time. During the leisure time, the four levels of physical activity correspond to the answer to the question “What is your physical activity during your leisure time?”:Usually reading, watching television, getting to the movies or spend leisure time in other sedentary activities (defined as sedentariness);Walking, riding a bicycle or to carry out some kind of physical activity for at least 4 h a week, anything more tiring than going to work on foot or by bicycle, gardening, hunting or fishing or playing ping-pong (defined as mild physical activity intensity);Doing sports as a hobby, such as running, swimming, tennis, gymnastics, or to do hard work in the garden or at home or other similar efforts (this is valid if this activity is carried out at least 3 times a week) (defined as moderate physical activity intensity);Training regularly or playing sport professionally such as athletics, skiing, swimming, football, basketball or tennis, several times a week (defined as heavy physical activity intensity).

Physical activity during leisure time was also assessed considering only participants who no longer carried out a working activity (retired). At work, the four levels of physical activity correspond to the answer to the question “What is (was) the physical activity deriving from your job?”:Work performed mainly sitting at a desk and generally without the need to walk; (defined as sedentariness);Work that implies standing and walking for a long time, but does not oblige to carry or move heavy weights (this category also includes normal housework, except hard work) (defined as mild physical activity intensity);Work that implies a lot of walking and handling heavy weights (this category also includes the cases of normal hard housework, such as doing the laundry, scrubbing the floors manually) (defined as moderate physical activity intensity);Hard manual work, with great efforts and lifting and handling heavy weights (defined as heavy physical activity intensity). Sedentariness was also investigated among retired participants, who no longer carries out a work activity. The questionnaire was previously used in an Italian research project sponsored by the National Research Council and in the previous Italian HESs within the CUORE Project [5,12].

### 2.4. Nutrition and Alcohol Consumption Data Collection

Dietary information was collected by the self-administered Italian version of the European Prospective Investigation into Cancer (EPIC) food frequency questionnaire (FFQ), which focused on diet as a major determinant of health. Accuracy and validation of the dietary questionnaire were of paramount importance since it had to be applied in several countries and to thousands of study participants [13]. The FFQ was designed to capture eating behaviours in the Italian population [14]. The questionnaire included images that defined food portions; it investigated general dietary habits (preferred food items, type of dressing, cooking modalities), frequency of meals consumed away from home and how frequently (weekly, monthly, yearly) each specific food was generally consumed.

For vegetables intake, the following foods were considered: leafy vegetables—cooked, leafy vegetables—raw, other vegetables, tomatoes—raw, tomatoes—cooked, root vegetables, cabbages, mushrooms, grain and pod vegetables, onion, garlic, stalk vegetables, mixed salad and mixed vegetables. For fruits intake, the following foods were considered: citrus fruits and other fruits. For fish intake, the following foods were considered: fish, crustaceans and molluscs. For cheese: cheeses (including fresh cheeses). For sweets/cakes intake, the following foods were considered: sugar, honey, jam, chocolate, candy bars, paste, confetti/flakes, non-chocolate confectionery, ice cream, cakes, pies, pastries, puddings (not milk-based), dry cakes and biscuits. For sweet beverages intake, the following foods were considered: fruit and vegetable juices, carbonated/soft/isotonic drinks and diluted syrups. For cereals intake, the following foods were considered: pasta, rice, white and wholemeal bread, other grains, crispbread, rusks and breakfast cereals. For potatoes intake, the following foods were considered: french fries, boiled potatoes, roasted potatoes, pure potatoes and croquette potatoes. For meats intake, the following foods were considered: beef, veal, pork, horse, chicken, turkey and rabbit (domestic).

A balanced nutrition was defined based on the following intake: vegetables ≥ 200 g/day; fruit 150–375 g/day; fish at least twice per week (150 g per serving); cheese no more than three times per week (50–100 g per serving); sausages, salami and other preserved meat no more than once per week (50 g per serving); cake and desserts no more than once per week (100 g per serving); sugar beverages less than once per week (330 mL); and consumption of alcoholic beverages limited to ≤24 g/day in men 35–64 years, ≤12 g/day in women 35–64 years and ≤12 g/day in men and women 65–74 years old [15,16,17,18,19]. These eight eating habits were also grouped together, and the population was divided into those who had no balanced eating behaviour or only one, and those who had 2, 3, 4, 5 or more balanced eating behaviours, respectively. Food items were linked using specifically designed software [20] to Italian Food Tables [21] to obtain estimates of daily intake of 37 macro- and micronutrients plus energy (not all nutrients are shown in the tables).

### 2.5. Statistical Analysis

The prevalence of people who followed a healthy lifestyle was evaluated, including them in the group of people who reported being non-smokers, non-sedentariness during leisure time and to have a food consumption comparable to at least five balanced eating behaviours. Educational level was used as a proxy for socioeconomic position; social class was characterized as those with primary/middle school degree (≤8 years)—lower education and those with high school/university degree (>8 years)—higher education. Data are presented separately for men and women, age classes, geographical areas and educational levels as mean, standard deviation and prevalence, with 95% confidence intervals (CI). Following the suggestion reported in the WHO Global NCDs Action Plan 2013–2020 extended to 2030 [1,2], indicators, where appropriate, were age standardized using the direct method, referring to the age- and sex-specific distributions of the Italian adult population in 2019 (Italian National Institute of Statistics—ISTAT) (Tables and figures, and Appendix A) [22]. Data were also age-standardized using the European Standard Population (EuStPop) 2013 for international comparisons (Appendix A) [23]. ANOVA was used to compare means and chi-square test to compare prevalence among classes. Two-sided *p*-values < 0.05 were considered statistically significant. Statistical analyses were performed using R software, release 4.2.3.

## 3. Results

From 2106 participants examined within the HES-CUORE Project 2018–2019, after excluding those with missing data for relevant variables, 2090 participants were considered in the statistical analyses for smoking habit and 2016 for physical activity; 1578 participants were considered in the statistical analyses for nutrition, after excluding those with missing data for relevant variables, those outside the range of the first and last percentile for the total energy consumption and the basal metabolic rate ratio (34 persons), and the participants residing in Abruzzo region with no nutritional data (211 persons).

### 3.1. Smoking Habit

Prevalence of current smokers of cigarettes was 23% in men and 19% in women, with a mean number of daily smoked cigarettes of 13 and 11, respectively, with significantly lower prevalence in older men (17%) in comparison to the first age-class, with no significant association by educational level and geographical area. In men, the mean number of cigarettes was higher in the older adults and in the Southern regions (Table 1, Appendix A).

Packaged cigarettes were consumed by 70% of smoking men and 84% of smoking women, handmade cigarettes by 19% and 7%, respectively, and electronic ones by 6%, both in smoking men and women. Prevalence of handmade and electronic cigarettes consumption decreased by age-class and among persons with a lower educational level (Figure 1, Appendix A).

Prevalence of former smokers of cigarettes was 43% in men and 28% in women, showing a significant increasing trend with age in both genders (from 24% of youngers to 47% of older men; from 19% to 30% in corresponding aged women); a decreasing trend with the educational level in men (54% in low educational level to 39% in high educational level), the opposite in women (24% and 30%, respectively). Prevalence of former-smoker men had a significant geographical gradient from Northern/Central Regions (40/41%) to Southern ones (47%); the opposite in women, from Northern/Central Regions (30/34%) to Southern ones (23%) (Table 1, Appendix A).

Cigar-smoker prevalence was 13% in men and 1% in women; 4% in men and 0.2% in women if in combination with cigarette smoking. Pipe-smoker prevalence was 7% in men and 1% in women; 2% in men and 0.3% in women in combination with cigarette smoking (Table 1 and Appendix A).

### 3.2. Physical Activity

During leisure time, the most prevalent activities were walking, cycling or engaging in physical activity of some kind for at least 4 h a week (43% of men and 40% of women), closely followed by sedentariness activities (34% and 45%), physical activity as a pastime for at least 3 h a week (20% and 14%) and systematic workout (4% and 2%) (Table 2, Appendix A, Figure 2 and Appendix A).

A significantly lower prevalence of sedentariness during leisure time was found in older men (22%) in comparison to the overall prevalence and first two age classes; it rises among retired older adults (37% in men and 56% in women) (Appendix A).

During working time, the highest prevalence resulted for sedentariness (45% in men and 47% in women), followed by standing or walking for a long time (38% and 42%, respectively), walking a lot and moving heavy weights (12% and 11%, respectively) and heavy manual work, with considerable effort (6% and 1%, respectively) (Table 2, Appendix A, Figure 2, Appendix A).

A significantly lower prevalence of sedentariness at work was found in persons with lower educational level in comparison to those with higher educational level (19% vs. 53% in men, 21% vs. 53% in women) (Table 2, Appendix A).

Combining physical activity during leisure time and at work in working people, 16% of men and 18% of women were classed as sedentary, with a significantly higher proportion in men and women with a higher educational level in comparison to those with a low educational level (18% vs. 8% in men, 21% vs. 9% in women) (Table 2, Appendix A).

### 3.3. Nutrition and Alcohol Consumption

Prevalence of balanced eating behaviours in men and women was, respectively: 28% and 39% for vegetables (a mean of 168 g per day in men and 184 g/day in women), 50% and 52% for fruits (a mean of 268 g/day in men and 276 g/day in women), cheese 45% and 54% (a mean of 43 g/day and 37 g/day, respectively), 11% and 21% for processed meats (36.4 g/day and 24.4 g/day), 44% and 40% for fish (46 g/day and 43 g/day), 9% and 11% for sweets/cakes (84 g/day and 78 g/day), 54% and 60%, for sweet drinks (100 mL/day and 73 mL/day) and 74% and 84% for alcohol consumption (15 mL/day and 6 mL/day) (Table 3, Table 4, Table 5, Table 6, Table 7 and Table 8 and Appendix A, Figure 3). 

Persons with healthy consumption of vegetables resulted tendentially lower among older adults (25% in men, 32% in women), in Southern regions (26% in men, 34% in women) and among persons with lower educational level (25% in men, 32% in women) (Table 3, Table 4, Appendix A, Figure 4 and Figure 5).

No associations were found between healthy consumption of fruit and class of age, geographical area and educational level (Table 3, Appendix A, Figure 4 and Figure 5). Prevalence of healthy consumption of cheese was tendentially higher in Southern regions (Table 3, Appendix A). Healthy consumption of processed meat was significantly lower among younger men and women (6% and 15%, respectively) and in lower educated men (7%) and higher educated women (20%) (Table 4 and Appendix A, Figure 4 and Figure 5). Healthy consumption of sweet drinks prevalence was significantly lower among younger men and women (41% and 46%, respectively) (Table 4, Appendix A, Figure 3). Prevalence of healthy consumption of alcohol significantly decreased by age-class, especially in men, and tendentially increased from Northern to Southern regions, but did not significantly differ by educational level (Table 4, Appendix A, Figure 3, Figure 4 and Figure 5). The most frequent number of balanced eating behaviours, out of the considered eight, was three both in men and women (about 30%); 12% of men and 23% of women presented with five or more. Prevalence of those with five or more balanced eating behaviours was lower in less-educated men, while similar prevalence was found in both high and low educational level for women (Figure 3, Figure 4 and Figure 5).

Regarding nutrients, significantly lower levels were found in men than in women for total protein (15% of total kcal in men and 16% of total kcal in women), total lipids (35% and 38% of total kcal, respectively), vegetable lipids (17% and 20% of total kcal, respectively), monounsaturated lipids (16% and 18% of total kcal, respectively), simple carbohydrates (19% and 20% of total kcal, respectively), cholesterol (154 and 162 mg/day, respectively); and fibre (10 and 11 g/day). Similar values were found in men and women for saturated fats (11% of total kcal for both sexes), carbohydrate (48% and 47% of total kcal, respectively), vegetables proteins (5% of total kcal for both sexes), animal protein (10% and 11% of total kcal, respectively), animal lipids (18%of total kcal for both sexes), polyunsaturated fat (5% of total kcal for both sexes) and potassium (3 g/day for both sexes). Statistically significant higher levels of sodium were found in men than in women (3 and 2 g/day, respectively) (Table 9, Table 10, Table 11 and Table 12, Appendix A).

Associations between nutrients and subgroups of population are worth mentioning: significantly lower levels were found in older men and women for total lipid, animal lipids, saturated fats, polyunsaturated fats and sodium; significant higher levels were found in older men and women for fibre and vegetables proteins; significant higher levels were found in men and women from Southern Regions for cholesterol, animal lipids and carbohydrate, lower levels for vegetables lipid; significant lower level were found in men and women from Northern Regions for fibre and vegetables proteins; significant higher levels were found in men and women with low educational level for cholesterol, vegetables proteins, vegetables lipids, lower levels for fibre (Table 9, Table 10, Table 11 and Table 12, Appendix A).

### 3.4. Healthy Lifestyles

Prevalence of people who followed a correct lifestyle (non-smoker, non-sedentary during leisure time and reporting at least five healthy eating behaviours) was 10% in men and 14% in women, with a higher prevalence in those with high educational level, especially in men (4% vs. 11% in men, 13% vs. 14% in women) (Figure 3, Figure 4 and Figure 5).

## 4. Discussion

Data on smoking habit collected within the Italian HES—CUORE Project in random samples of the general Italian population aged 35–74 years during 2018–2019 showed about a fifth of the population as current smokers of cigarettes, with consistently higher levels for men than for women in all age-groups, geographical areas and educational levels; the same was found for the number of smoked cigarettes among smokers. Over 60% of men and about 50% of women are or have been habitual smokers. Regardless of gender, geographical area or level of education, the most consumed type of cigarette remains the packaged, followed by the handmade, with a prevalence of use of electronic cigarette not higher than 10%.

Data on physical activity collected within the Italian HES—CUORE Project during 2018–2019 showed about a third of men and almost half of women conducting no physical activity in their free time, with peaks of 66% in women living in a Southern region. Physical inactivity at work was around 46% in both men and women; it exceeds 50% in people with a higher educational level versus around 20% in people with a lower educational level. Just less than 2 in 10 people live in a sedentary condition both at work and during leisure time. Among retirees, more than a third of men and half of women follow a sedentary lifestyle.

Data on nutrition collected within the Italian HES 2018–2019—CUORE Project showed that women more frequently than men had a number of balanced eating behaviours equal to or greater than four (out of eight). More than two-thirds of men and women consumed alcohol within the limits of consumption in relation to the gender and age; around 4/5 in 10 men and women consume amounts of fruit, fish, cheese and sugary drinks as recommended by the Italian guidelines for an healthy nutrition [15]; less than 3 in 10 men and about 3 in 10 women consume the right amount of vegetables, while the correct consumption of processed meat does not exceed 1 in 10 people in men and 2 in women; 1 in 10 also met the recommended consumption of sweets/cakes both in men and women. There was a greater propensity in people with a higher educational level, especially men, to have healthy consumption of vegetables, processed meat and sweets/cakes, which represent, among the food groups considered (vegetables, fruits, fish, cheeses, processed meat, sweets/cakes, sweet drinks and alcohol), the food groups with the lowest prevalence of balanced eating behaviours.

Regarding the older adults, a better lifestyle profile was found compared to the general population, in fact the results showed tendentially lower prevalence of smoking habit and a higher prevalence of former smokers, significantly lower prevalence of sedentariness during leisure time, significantly higher prevalence of healthy consumption of processed meat, sweet drinks, alcohol, significantly lower levels of total lipid, animal lipids, saturated fats, polyunsaturated fats and sodium and significantly higher levels of for fibre and vegetables proteins.

In comparison to previous HES conducted at national level in Italy, prevalence of current smokers 2018–2019 decreased compared to 20 years before (1998–2002) and very slightly declined compared to 10 years before (2008–2012), both in men and women, while former smokers increased [5,24]. The trend of current-smoker prevalence is consistent with WHO data that showed the age-standardized tobacco use prevalence rates are declining, on average, in all WHO regions; in particular, in the European region, a relatively slow decline is being recorded [25]; prevalence and trends are also consistent with other Italian studies [26,27]. A 30% relative reduction in prevalence of current tobacco use within 2025, extended to 40% within 2030 (WHO targets for the prevention of NCDs), in Italy means a reduction in absolute terms of 7% in men and 6% in women within 2025, considering the 2008–2012 smoking prevalence as the reference (24% and 20% of cigarettes smokers in men and in women); reductions very far from the approximately 1% and 1.5% decline resulted in 2019 in men and in women, respectively [1,2,5,24].

Compared to 10 years before (2008–2012), the 2018–2019 prevalence of a sedentariness lifestyle during leisure time increased, both in men and women, returning to around the values found 20 years before (1998–2002) and showing an increase in the gap between levels of education in both sexes [5]. Prevalence of sedentariness from the HES 2018–2019 was in line with the European results showing that more than one third of adults are insufficiently active [28], with an increasing trend in high-income Western countries [29] and with the propensity of women to be less active than men [28,29]. A 10% relative reduction in prevalence of sedentariness lifestyle within 2025, extended to 15% within 2030 (WHO targets for the prevention of NCDs), in Italy means a reduction in absolute terms of 3% in men and 4% in women within 2025, considering the 2008–2012 prevalence of sedentariness during leisure time as the reference (32% and 42% of sedentariness in men and women). Taking into account the observed trend until 2019, a stable reversal trend will represent a first goal [1,2,5,24].

Compared to 10 years before (2008–2012), the prevalence of balanced eating behaviours in 2018–2019 in Italy improved for vegetables and sweets/cakes in women, and for fish, sweet drinks and alcohol in both sexes; but got worse for fruits and processed meat in both men and women, and sweets/cakes in men [5]. Mean intake of food groups was less than desirable, particularly for vegetables and fibres [30]. Mean intake of nutrients resulted in desirable intervals, except for an excessive consumption of total lipid, both saturated fat and polyunsaturated fat, simple carbohydrates and not enough intake of fibre and potassium; mean level of nutrients remains stable, except for a decreased cholesterol and fibre intake, both in men and women [31]. Mean values of sodium intake were significantly lower than those assessed in the same samples through the 24 h collection through which excessive consumption was found [8]; this result was due to the fact that questionnaires are not able to register the discretionary salt added during the preparation of meals and at table, which is instead included in the measurement carried out by the urine collection method; a similar order of difference was found for potassium consumption between the two methods of collection [9]. There was no homogeneity in fruit and vegetable consumption across European countries: fruit consumption increased from 1950 to 2019 in Northern and Western Europe, while the greatest increase in vegetable consumption occurred in the Middle East and Northern Europe, followed by Western Europe [32,33]. 

About the alcohol consumption, it is worth specifying that although protective associations have been reported in the literature between occasional consumption of alcohol and coronary heart disease, ischemic stroke and diabetes, according to WHO, net of all supposed benefits, the harmful effects of alcohol consumption on health are nonetheless preponderant [34,35]. In Italy, the guidelines for low-risk consumption have been included in the Scientific Dossier of the Guidelines for balanced eating [36,37], edited by the Council for Research in Agriculture and Agricultural Economics (CREA), reiterating that there are no safe levels of alcohol consumption. The non-“excess habitual consumption” is considered here as “healthy” way of consuming alcoholic beverages considering the limits of alcoholic beverages consumption in relation to the gender and age of the person [16,17,18,19].

Overall, the prevalence of healthy lifestyles assessed in the HES 2018–2019 was higher than 10 years earlier (2008–2012) both in men and women; in men the prevalence increased for both educational level, in women in those with lower educational level. The prevalence of people with at least five correct eating habits was higher than 10 years earlier (2008–2012), especially among men with higher education and women with lower education; a slight decreasing trend of people with only one or less correct eating habits was found [5,25].

Major strengths of this study are the following: the use of standardized questionnaires to collect very detailed data on smoking habit, physical activity, and, in a particular way, on nutrition; a good national coverage with the enrolment of study participants from half of the Italian regions distributed in the northern, central and southern areas of the country; the use of randomly selected samples of the general population stratified by sex and age-group. Conversely, we acknowledge some study limitations, which should be taken in consideration when interpreting results. First, because of the choice of urban districts for the random selection of the study participants, the results may not be representative of the population living in rural areas. The participation rates in the surveys were lower than desirable, yet consistent, taking into account the low contact rates occurring in more highly urbanized areas and the decreasing trend of participation observed in the HESs implemented in other European countries [38].

## 5. Conclusions

Data from some main lifestyle factors such as smoking habit, physical activity and nutrition collected within the Italian HES—CUORE Project in random samples of the general Italian population aged 35–74 years during 2018–2019 still showed epidemic levels of prevalence of current smokers, physical inactivity and incorrect eating behaviours, with an increasing prevalence of combined healthy lifestyles (not smoker, engaged in some regular physical activity and reporting at least five correct eating behaviours) which remains more prevalent in women. The prevalence of individual lifestyles is overall stable with small variations: slight decrease in smoking habit, slight increase in sedentariness during leisure time, slight increase in healthy consumption of vegetable and fish and slight decrease in healthy consumption of processed meat. The overall Italian picture of slight improvements in some healthy lifestyle factors and in the prevalence of combined correct lifestyles, together with the reduction in blood pressure and salt consumption, as well as the stop to the increasing obesity detected in the HES 2018–2019 as compared to the previous 10 years in the Italian general adult population, can be considered a partial success [6,7,8]. National and local cross-sectoral strategies, based on actions that involve different sectors of society and institutions, and specific interventions to fight smoking and alcohol abuse and to promote physical activity and balanced eating should be continued, as well as the monitoring of lifestyles and related risk factors and conditions.

## Figures and Tables

**Figure 1 healthcare-12-00475-f001:**
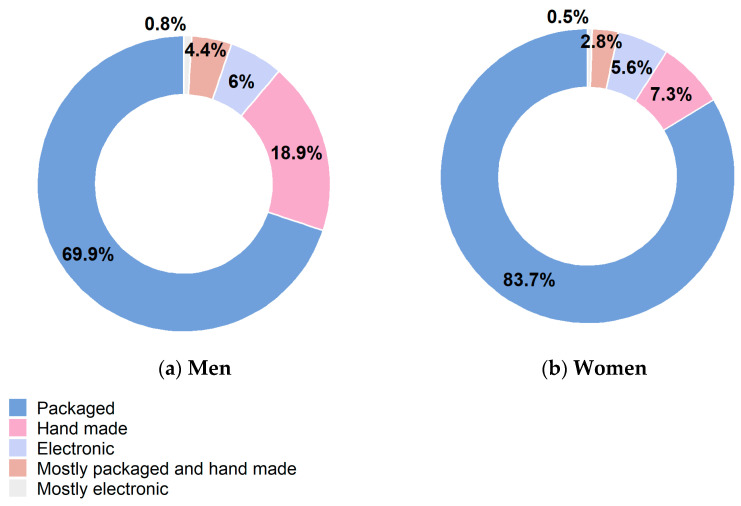
Age-standardized (Italian population) type of daily consumed cigarettes by sex. Men and women residing in Italy aged 35–74 years, Health Examination Survey 2018–2019—CUORE Project. The pool was made of the following Italian regions: Piedmont, Lombardy, Liguria, Emilia Romagna, Tuscany, Lazio, Abruzzo, Basilicata, Calabria and Sicily. Prevalence was age standardized by Italian National Institute of Statistics—ISTAT Italian population 2019 (except when it is reported by age-classes). (**a**) Men; (**b**) Women.

**Figure 2 healthcare-12-00475-f002:**
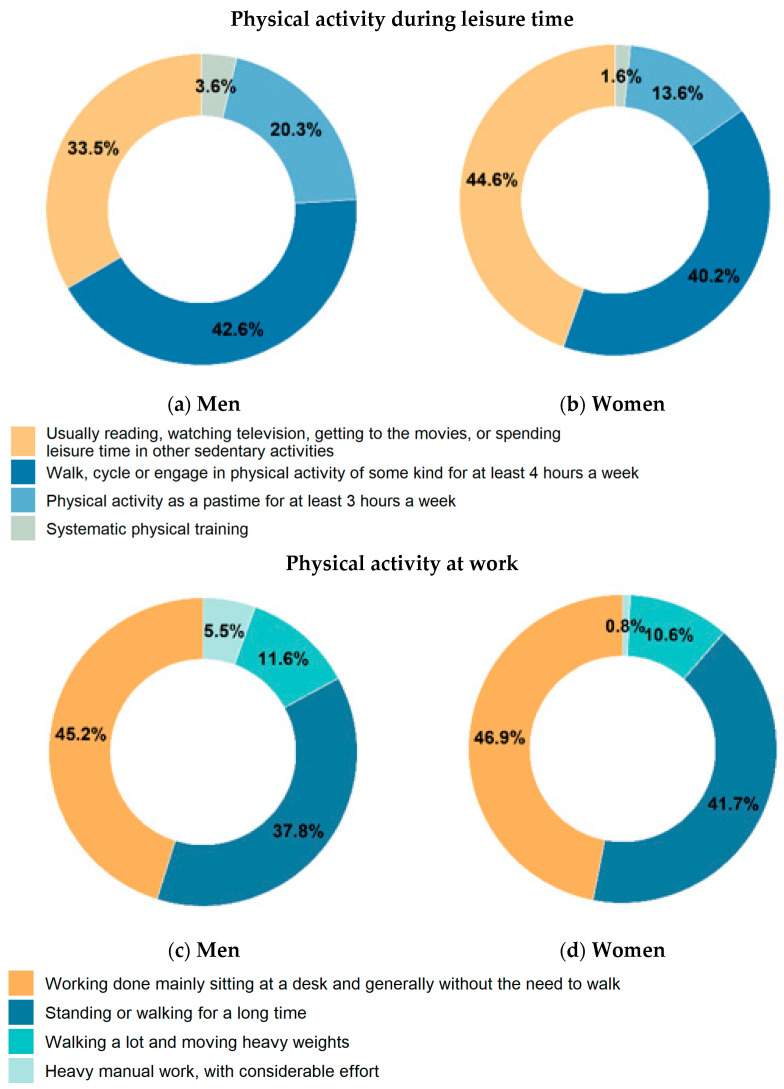
Age-standardized (Italian population) physical activity during leisure time and at work by sex. Men and women residing in Italy aged 35–74 years, Health Examination Survey 2018–2019—CUORE Project. The pool was made of the following Italian regions: Piedmont, Lombardy, Liguria, Emilia Romagna, Tuscany, Lazio, Abruzzo, Basilicata, Calabria and Sicily. Prevalence was age standardized by Italian National Institute of Statistics—ISTAT Italian population 2019 (except when it is reported by age-classes). (**a**) Men (**b**) Women (**c**) Men (**d**) Women.

**Figure 3 healthcare-12-00475-f003:**
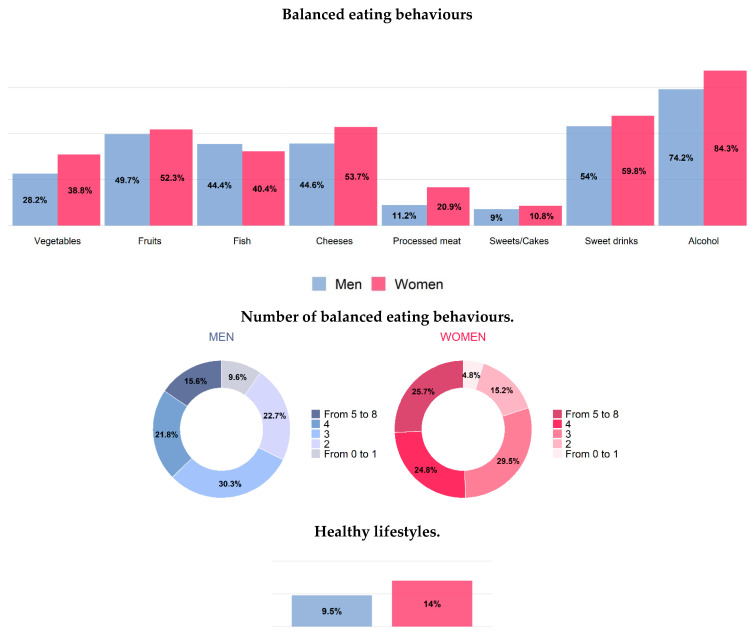
Age-standardized (Italian population) nutrition and healthy lifestyles (EPIC questionnaire) by sex. Men and women residing in Italy aged 35–74 years, Health Examination Survey 2018–2019—CUORE Project. For vegetables intake, the following foods were considered: leafy vegetables—cooked, leafy vegetables—raw, other vegetables, tomatoes—raw, tomatoes—cooked, root vegetables, cabbages, mushrooms, grain and pod vegetables, onion, garlic, stalk vegetables, mixed salad, mixed vegetables. For fruits intake, the following foods were considered: citrus fruits and other fruits. For fish intake, the following foods were considered: fish, crustaceans and molluscs. For cheese intake, the following foods were considered: cheeses (including fresh cheeses). For processed meat intake, the following foods were considered: sausages, salami and other preserved meat. For sweets/cakes intake, the following foods were considered: sugar, honey, jam, chocolate, candy bars, paste, confetti/flakes, non-chocolate confectionery, ice cream, cakes, pies, pastries, puddings (not milk-based), dry cakes and biscuits. For sweet beverages intake, the following foods were considered: fruit and vegetable juices, carbonated/soft/isotonic drinks and diluted syrups. For alcohol intake, the following foods were considered: alcoholic beverages. These eight eating habits were also grouped together, and the population was divided into those who had no healthy eating behaviour or only one, and those who had, respectively, 2, 3, 4, 5 or more healthy eating behaviours. A balanced nutrition was defined based on the following intake: vegetables ≥ 200 g/day; fruit 150–375 g/day; fish at least twice per week (150 g per serving); cheese no more than three times per week (50–100 g per serving); sausages, salami and other preserved meat no more than once per week (50 g per serving); cake and desserts no more than once per week (100 g per serving); sugar beverages less than one per week (330 mL); and consumption of alcoholic beverages limited to two glasses per day for men (24 g of ethanol), one glass per day for women (12 g of ethanol) and one glass per day for men and women aged 65–74 (12 g of ethanol). Healthy lifestyle: people who reported being non-smokers, not sedentary during leisure time and to have a food consumption comparable to at least five balanced eating behaviours. The pool was made of the following Italian regions: Piedmont, Lombardy, Liguria, Emilia Romagna, Tuscany, Lazio, Basilicata, Calabria and Sicily. Prevalence was age standardized by Italian National Institute of Statistics—ISTAT Italian population 2019 (except when it is reported by age-classes).

**Figure 4 healthcare-12-00475-f004:**
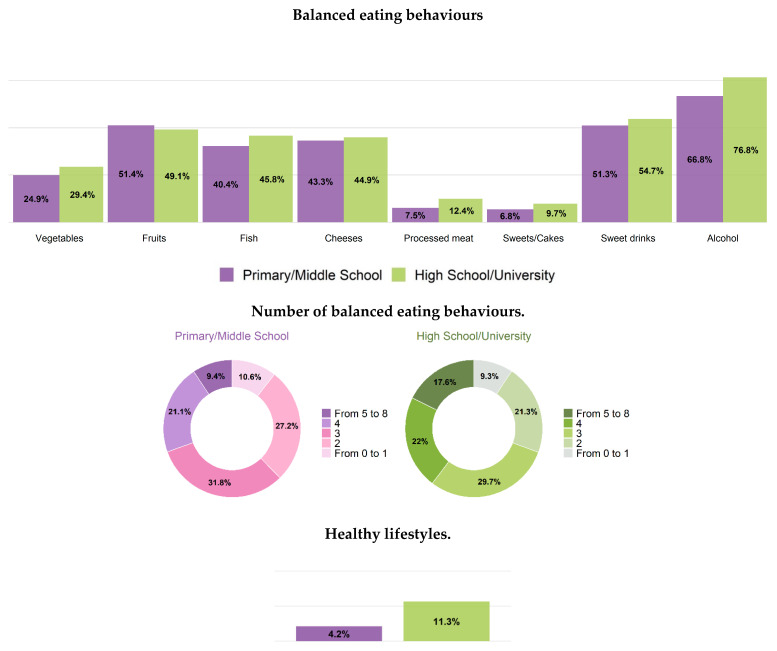
Age-standardized (Italian population) nutrition and healthy lifestyles (EPIC questionnaire) by education level. Men residing in Italy aged 35–74 years, Health Examination Survey 2018–2019—CUORE Project. For vegetables intake, the following foods were considered: leafy vegetables—cooked, leafy vegetables—raw, other vegetables, tomatoes—raw, tomatoes—cooked, root vegetables, cabbages, mushrooms, grain and pod vegetables, onion, garlic, stalk vegetables, mixed salad and mixed vegetables. For fruits intake, the following foods were considered: citrus fruits and other fruits. For fish intake, the following foods were considered: fish, crustaceans and molluscs. For cheese intake, the following foods were considered: cheeses (including fresh cheeses). For processed meat intake, the following foods were considered: sausages, salami and other preserved meat. For sweets/cakes intake, the following foods were considered: sugar, honey, jam, chocolate, candy bars, paste, confetti/flakes, non-chocolate confectionery, ice cream, cakes, pies, pastries, puddings (not milk-based), dry cakes and biscuits. For sweet beverages intake, the following foods were considered: fruit and vegetable juices, carbonated/soft/isotonic drinks and diluted syrups. For alcohol intake, the following foods were considered: alcoholic beverages. These eight eating habits were also grouped together and the population was divided into those who had no healthy eating behaviour or only one, and those who had, respectively, 2, 3, 4, 5 or more healthy eating behaviours. A balanced nutrition was defined based on the following intake: vegetables ≥ 200 g/day; fruit 150–375 g/day; fish at least twice per week (150 g per serving); cheese no more than three times per week (50–100 g per serving); sausages, salami and other preserved meat no more than once per week (50 g per serving); cake and desserts no more than once per week (100 g per serving); sugar beverages less than one per week (330 mL); and consumption of alcoholic beverages limited to two glasses per day for men (24 g of ethanol), one glass per day for women (12 g of ethanol) and one glass per day for men and women aged 65–74 (12 g of ethanol). Healthy lifestyle: people who reported not being smoker, not sedentariness during leisure time, and to have a food consumption comparable to at least five balanced eating behaviours. The pool was made of the following Italian regions: Piedmont, Lombardy, Liguria, Emilia Romagna, Tuscany, Lazio, Basilicata, Calabria and Sicily. Prevalence was age standardized by Italian National Institute of Statistics—ISTAT Italian population 2019 (except when it is reported by age-classes).

**Figure 5 healthcare-12-00475-f005:**
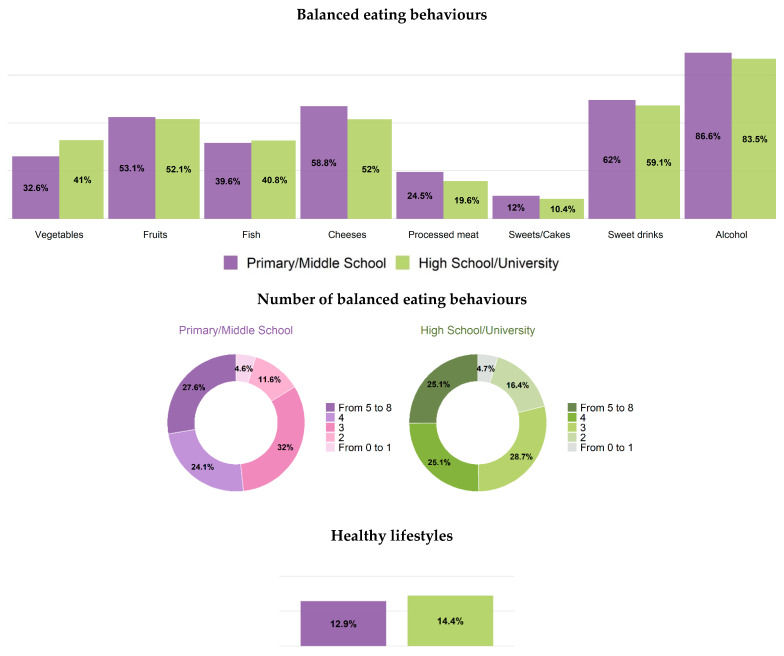
Age-standardized (Italian population) nutrition and healthy lifestyles (EPIC questionnaire) by education level. Women residing in Italy aged 35–74 years, Health Examination Survey 2018–2019—CUORE Project. For vegetables intake, the following foods were considered: leafy vegetables—cooked, leafy vegetables—raw, other vegetables, tomatoes—raw, tomatoes—cooked, root vegetables, cabbages, mushrooms, grain and pod vegetables, onion, garlic, stalk vegetables, mixed salad, mixed vegetables. For fruits intake, the following foods were considered: citrus fruits and other fruits. For fish intake, the following foods were considered: fish, crustaceans and molluscs. For cheese intake, the following foods were considered: cheeses (including fresh cheeses). For processed meat intake, the following foods were considered: sausages, salami and other preserved meat. For sweets/cakes intake, the following foods were considered: sugar, honey, jam, chocolate, candy bars, paste, confetti/flakes, non-chocolate confectionery, ice cream, cakes, pies, pastries, puddings (not milk-based), dry cakes and biscuits. For sweet beverages intake, the following foods were considered: fruit and vegetable juices, carbonated/soft/isotonic drinks and diluted syrups. For alcohol intake, the following foods were considered: alcoholic beverages. These eight eating habits were also grouped together, and the population was divided into those who had no healthy eating behaviour or only one, and those who had, respectively, 2, 3, 4, 5 or more healthy eating behaviours. A balanced nutrition was defined based on the following intake: vegetables ≥ 200 g/day; fruit 150–375 g/day; fish at least twice per week (150 g per serving); cheese no more than three times per week (50–100 g per serving); sausages, salami and other preserved meat no more than once per week (50 g per serving); cake and desserts no more than once per week (100 g per serving); sugar beverages less than one per week (330 mL); and consumption of alcoholic beverages limited to two glasses per day for men (24 g of ethanol), one glass per day for women (12 g of ethanol) and one glass per day for men and women aged 65–74 (12 g of ethanol). Healthy lifestyle: people who reported not being smoker, not sedentariness during leisure time, and to have a food consumption comparable to at least five balanced eating behaviours. The pool was made of the following Italian regions: Piedmont, Lombardy, Liguria, Emilia Romagna, Tuscany, Lazio, Basilicata, Calabria and Sicily. Prevalence was age-standardized by the Italian National Institute of Statistics—ISTAT Italian population 2019 (except when it is reported by age-classes).

**Table 1 healthcare-12-00475-t001:** Age-standardized (Italian population) current smokers and former smokers prevalence, and mean number of cigarettes smoked per day, by sex, age classes, geographical area and educational level. Cigarette, cigar and pipe smokers prevalence. Men and women residing in Italy aged 35–74 years, Health Examination Survey 2018–2019—CUORE Project.

	Men						Women					
	**Current Smokers**											
	**N**	**n**	**%**	**CI 95%**		**χ^2^ ** ** *p* ** **-value**	**N**	**n**	**%**	**CI 95%**		**χ^2^ ** ** *p* ** **-value**
**All**	**1031**	**234**	**23.1**	**18.0**	**28.3**		**1059**	**196**	**18.6**	**13.9**	**23.3**	
**Age classes (years)**												
35–44	249	72	28.9	23.3	34.6	0.0070	235	49	20.9	15.7	26.1	0.2763
45–54	248	63	25.4	20.0	30.8		271	48	17.7	13.2	22.3	
55–64	276	54	19.6	14.9	24.3		276	57	20.7	15.9	25.4	
65–74	258	45	17.4	12.8	22.1		277	42	15.2	10.9	19.4	
**Italian Area**												
North	409	88	21.5	13.6	29.5	0.1115	421	74	17.5	10.2	24.8	0.8172
Centre	206	58	29.4	16.9	41.8		208	40	19.5	8.7	30.3	
South	416	88	21.6	13.6	29.5		430	82	19.3	11.8	26.7	
**Education**												
Primary and middle school	303	72	25.0	15.0	34.9	0.6556	338	65	19.8	11.1	28.4	0.7491
High school and university	728	162	22.4	16.4	28.4		720	131	18.1	12.5	23.7	
	**Daily smoke—number of cigarettes**											
	**N**	**Mean**	**SD**	**CI 95%**		**ANOVA ** ** *p* ** **-value**	**N**	**Mean**	**SD**	**CI 95%**		**ANOVA ** ** *p* ** **-value**
**All**	**234**	**13.3**	**8.9**	**12.8**	**13.9**		**196**	**10.8**	**6.4**	**10.4**	**11.2**	
**Age classes (years)**												
35–44	72	12.3	8.4	11.3	13.4	0.0155	49	8.2	5.7	7.5	8.9	0.0155
45–54	63	14.1	9.9	12.9	15.3		48	12.0	7.3	11.1	12.8	
55–64	54	11.6	6.6	10.8	12.4		57	11.2	5.8	10.5	11.9	
65–74	45	16.3	10.2	15.0	17.5		42	11.4	5.9	10.7	12.1	
**Italian Area**												
North	88	12.8	9.9	11.9	13.8	0.573	74	10.8	7.8	10.0	11.5	0.573
Centre	58	12.3	7.6	11.2	13.3		40	10.7	5.6	10.0	11.5	
South	88	14.6	8.6	13.7	15.4		82	10.8	5.4	10.3	11.3	
**Education**												
Primary and middle school	72	16.6	10.0	15.5	17.7	<0.0001	65	12.8	7.0	12.1	13.6	<0.0001
High school and university	162	11.9	7.9	11.3	12.4		131	9.8	5.8	9.3	10.2	
**Former Smokers**												
	**N**	**n**	**%**	**CI 95%**		**χ^2^ ** ** *p* ** **-value**	**N**	**n**	**%**	**CI 95%**		**χ^2^ ** ** *p* ** **-value**
**All**	**1031**	**351**	**42.9**	**36.8**	**48.9**		**1059**	**246**	**27.8**	**22.4**	**33.2**	
**Age classes (years)**												
35–44	249	59	23.7	18.4	29.0	<0.0001	235	44	18.7	13.7	23.7	<0.0001
45–54	248	68	27.4	21.9	33.0		271	42	15.5	11.2	19.8	
55–64	276	102	37.0	31.3	42.7		276	77	27.9	22.6	33.2	
65–74	258	122	47.3	41.2	53.4		277	83	30.0	24.6	35.4	
**Italian Area**												
North	409	130	39.9	30.4	49.4	0.0851	421	105	29.7	21.0	38.5	0.0148
Centre	206	61	40.9	27.4	54.3		208	59	34.0	21.0	46.9	
South	416	160	46.8	37.1	56.4		430	82	23.0	15.1	31.0	
**Education**												
Primary and middle school	303	127	53.9	42.5	65.4	0.0001	338	65	23.6	14.4	32.8	0.0458
High school and university	728	224	38.6	31.6	45.7		720	181	29.7	23.1	36.3	
	**Cigar Smokers**											
	**N**	**n**	**%**	**CI 95%**		**χ^2^ ** ** *p* ** **-value**	**N**	**n**	**%**	**CI 95%**		**χ^2^ ** ** *p* ** **-value**
**All**	**1031**	**115**	**13**	**8.9**	**17.1**	**<0.0001**	**1059**	**10**	**1.2**	**0**	**2.5**	**<0.0001**
	**Pipe Smokers**											
	**N**	**n**	**%**	**CI 95%**		**χ^2^** ***p*-value**	**N**	**n**	**%**	**CI 95%**		**χ^2^** ***p*-value**
**All**	**1031**	**68**	**7.4**	**4.2**	**10.6**	**<0.0001**	**1059**	**7**	**0.8**	**0**	**1.8**	**<0.0001**
	**Cigarette and Cigar Smokers**											
	**N**	**n**	**%**	**CI 95%**		**χ^2^** ***p*-value**	**N**	**n**	**%**	**CI 95%**		**χ^2^** ***p*-value**
**All**	**1031**	**38**	**3.7**	**1.4**	**6**	**<0.0001**	**1059**	**2**	**0.2**	**0**	**0.8**	**<0.0001**
	**Cigarette and Pipe Smokers**											
	**N**	**n**	**%**	**CI 95%**		**χ^2^** ***p*-value**	**N**	**n**	**%**	**CI 95%**		**χ^2^** ***p*-value**
**All**	**1031**	**24**	**2.2**	**0.4**	**4.1**	**<0.0001**	**1059**	**3**	**0.3**	**0**	**0.9**	**<0.0001**

N: number of participants denominator. n: number of participants related to the prevalence numerator. Current smoker is defined as a person who smokes one or more cigarettes per day. A former smoker is defined as a person who has quit smoking for at least 12 months. Number of cigarettes refers to packaged, handmade or electronic cigarettes. SD: standard deviation; CI: confidence interval. Means, standard deviations and prevalence were age standardized by Italian National Institute of Statistics—ISTAT Italian population 2019 (except when they are reported by age-classes). ANOVA to compare mean values among classes; chi-square test to compare prevalence among classes. The pool was made of the following Italian regions: Piedmont, Lombardy, Liguria, Emilia Romagna, Tuscany, Lazio, Abruzzo, Basilicata, Calabria and Sicily. Italian Area: North (Piedmont, Lombardy, Liguria, Emilia Romagna); Centre (Tuscany, Lazio); South (Abruzzo, Basilicata, Calabria, Sicily).

**Table 2 healthcare-12-00475-t002:** Age-standardized (Italian population) sedentariness during leisure time and at work prevalence by sex, age classes, geographical area and educational level. Men and women residing in Italy aged 35–74 years, Health Examination Survey 2018–2019—CUORE Project.

	Men						Women					
	**Sedentariness during Leisure Time**											
	**N**	**n**	**%**	**CI 95%**		**χ^2^ ** ** *p* ** **-value**	**N**	**n**	**%**	**CI 95%**		**χ^2^ ** ** *p* ** **-value**
**All**	**1035**	**335**	**33.5**	**27.7**	**39.2**		**1071**	**474**	**44.6**	**38.7**	**50.6**	
**Age classes (years)**												
35–44	249	92	37.0	31.0	42.9	<0.0001	238	102	42.9	36.6	49.1	0.6034
45–54	249	101	40.6	34.5	46.7		272	130	47.8	41.9	53.7	
55–64	277	84	30.3	24.9	35.7		282	124	44.0	38.2	49.8	
65–74	260	58	22.3	17.3	27.4		279	118	42.3	36.5	48.1	
**Italian Area**												
North	410	124	31.0	22.1	40.0	0.1783	426	155	37.0	27.8	46.1	<0.0001
Centre	207	62	30.8	18.2	43.4		213	91	42.8	29.5	56.2	
South	418	149	37.2	27.9	46.5		432	228	53.1	43.7	62.5	
**Education**												
Primary and middle school	304	115	39.4	28.1	50.6	<0.0001	341	182	53.8	43.0	64.5	0.5837
High school and university	728	220	31.1	24.5	37.8		728	291	40.5	33.4	47.6	
	**Sedentariness at work**											
	**N**	**n**	**%**	**CI 95%**		**χ^2^ ** ** *p* ** **-value**	**N**	**n**	**%**	**CI 95%**		**χ^2^ ** ** *p* ** **-value**
**All**	**767**	**332**	**45.2**	**38.3**	**52.0**		**833**	**290**	**46.9**	**40.2**	**53.6**	
**Age classes (years)**												
35–44	240	89	38.4	26.1	50.6	0.0125	231	92	47.4	34.3	60.6	0.9172
45–54	244	104	44.1	32.7	55.5		265	94	45.2	34.1	56.3	
55–64	227	117	53.7	40.6	66.7		244	92	48.7	36.2	61.2	
65–74	56	22	46.8	17.1	76.5		93	12	46.2	23.7	68.6	
**Italian Area**												
North	315	140	45.2	34.5	55.9	0.6505	314	128	50.9	40.0	61.8	0.0557
Centre	159	73	47.7	32.5	62.9		166	66	50.6	35.5	65.6	
South	293	119	43.7	32.6	54.8		353	96	40.6	30.5	50.7	
**Education**												
Primary and middle school	175	31	19.0	7.7	30.4	<0.0001	231	25	21.0	10.5	31.5	<0.0001
High school and university	589	301	52.7	44.8	60.5		600	265	53.1	45.3	61.0	
	**Sedentariness during leisure time and at work**											
	**N**	**n**	**%**	**CI 95%**		**χ^2^** ***p*-value**	**N**	**n**	**%**	**CI 95%**		**χ^2^** ***p*-value**
**All**	**752**	**114**	**15.6**	**10.5**	**20.6**		**734**	**132**	**18.2**	**13.0**	**23.3**	
**Age classes (years)**												
35–44	234	30	12.8	4.4	21.3	0.04	213	41	19.3	8.9	29.6	0.0940
45–54	241	47	19.5	10.4	28.6		237	44	18.6	9.9	27.2	
55–64	224	34	15.2	5.8	24.6		221	43	19.5	9.6	29.3	
65–74	53	3	5.7	0.0	19.4		63	4	6.4	0.0	17.2	
**Italian Area**												
North	312	41	13.4	6.1	20.8	0.3834	296	49	17.1	8.9	25.3	0.7056
Centre	155	24	15.7	4.6	26.7		151	29	19.3	7.4	31.1	
South	285	49	17.8	9.2	26.4		287	54	18.7	10.7	26.7	
**Education**												
Primary and middle school	169	14	8.2	0.2	16.1	0.0068	172	15	8.9	1.6	16.3	0.0005
High school and university	583	100	17.7	11.7	23.7		562	117	20.9	14.5	27.3	

N: number of participants denominator. n: number of participants related to the prevalence numerator. Sedentariness during leisure time is defined as someone who usually carries out activities such as reading, watching television, getting to the movies or spend leisure time in other sedentary activities. Sedentariness at work is defined as one whose work mainly involves sitting at a desk and generally without the need to walk. SD: standard deviation; CI: confidence interval. Prevalence was age standardized by Italian National Institute of Statistics—ISTAT Italian population 2019 (except when it is reported by age-classes). Chi-square test to compare prevalence among classes. The pool was made of the following Italian regions: Piedmont, Lombardy, Liguria, Emilia Romagna, Tuscany, Lazio, Abruzzo, Basilicata, Calabria and Sicily. Italian Area: North (Piedmont, Lombardy, Liguria, Emilia Romagna); Centre (Tuscany, Lazio); South (Abruzzo, Basilicata, Calabria, Sicily).

**Table 3 healthcare-12-00475-t003:** Age-standardized (Italian population) balanced nutrition food groups intake (EPIC questionnaire): vegetables, fruit, fish and cheese prevalence by sex, age classes, geographical area and educational level. Men and women residing in Italy aged 35–74 years, Health Examination Survey 2018–2019—CUORE Project.

	Men						Women					
	**Vegetables**											
	**N**	**n**	**%**	**CI 95%**		**χ^2^** ***p*-value**	**N**	**n**	**%**	**CI 95%**		**χ^2^** ***p*-value**
**All**	**793**	**222**	**28.2**	**22.0**	**34.5**		**785**	**302**	**38.8**	**32.0**	**45.6**	
**Age classes (years)**												
35–44	196	51	26.0	13.8	38.3	0.4916	191	86	45.0	30.6	59.4	0.0301
45–54	204	61	29.9	18.4	41.4		216	90	41.7	29.5	53.8	
55–64	219	67	30.6	18.3	42.9		214	74	34.6	21.9	47.3	
65–74	174	43	24.7	10.3	39.2		164	52	31.7	16.2	47.3	
**Italian Area**												
North	357	97	27.3	18.1	36.6	0.2484	359	140	39.4	29.3	49.5	0.2616
Centre	180	59	33.0	19.2	46.7		184	78	43.2	28.9	57.5	
South	256	66	26.2	15.5	36.9		242	84	34.5	22.6	46.3	
**Education**												
Primary and middle school	201	50	24.9	12.8	37.0	0.2883	209	68	32.6	19.8	45.4	0.0463
High school and university	591	172	29.4	22.1	36.7		575	234	41.0	33.1	49.0	
	**Fruit**											
	**N**	**n**	**%**	**CI 95%**		**χ^2^** ***p*-value**	**N**	**n**	**%**	**CI 95%**		**χ^2^** ***p*-value**
**All**	**793**	**394**	**49.7**	**42.8**	**56.7**		**785**	**411**	**52.3**	**45.4**	**59.3**	
**Age classes (years)**												
35–44	196	96	49.0	42.0	56.0	0.6598	191	98	51.3	44.2	58.4	0.9483
45–54	204	106	52.0	45.1	58.8		216	111	51.4	44.7	58.1	
55–64	219	102	46.6	40.0	53.2		214	115	53.7	47.1	60.4	
65–74	174	90	51.7	44.3	59.2		164	87	53.1	45.4	60.7	
**Italian Area**												
North	357	179	50.0	39.6	60.3	0.2694	359	196	55.3	45.1	65.6	0.4259
Centre	180	97	53.5	38.9	68.1		184	96	51.7	37.3	66.2	
South	256	118	46.7	34.5	58.9		242	119	48.3	35.8	60.8	
**Education**												
Primary and middle school	201	104	51.4	37.4	65.4	0.5390	209	111	53.1	39.5	66.8	0.8798
High school and university	591	289	49.1	41.1	57.1		575	300	52.1	44.0	60.2	
	**Vegetable and Fruit**											
	**N**	**n**	**%**	**CI 95%**		**χ^2^** ***p*-value**	**N**	**n**	**%**	**CI 95%**		**χ^2^** ***p*-value**
**All**	**793**	**104**	**13.5**	**8.7**	**18.2**		**785**	**154**	**19.9**	**14.3**	**25.5**	
**Age classes (years)**												
35–44	196	26	13.3	3.8	22.7	0.2520	191	39	20.4	8.8	32.1	0.3179
45–54	204	34	16.7	7.3	26.0		216	50	23.2	12.7	33.6	
55–64	219	27	12.3	3.6	21.1		214	39	18.2	7.9	28.5	
65–74	174	17	9.8	0.0	19.7		164	26	15.9	3.7	28.1	
**Italian Area**												
North	357	50	14.1	6.9	21.4	0.0678	359	82	23.4	14.7	32.1	0.1014
Centre	180	30	17.0	6.0	28.0		184	33	18.4	7.2	29.6	
South	256	24	10.0	2.7	17.4		242	39	15.9	6.8	25.0	
**Education**												
Primary and middle school	201	22	10.9	2.2	19.7	0.3465	209	35	17.0	6.8	27.3	0.2589
High school and university	591	82	14.3	8.7	19.9		575	119	20.9	14.3	27.5	
	**Fish**											
	**N**	**n**	**%**	**CI 95%**		**χ^2^** ***p*-value**	**N**	**n**	**%**	**CI 95%**		**χ^2^** ***p*-value**
**All**	**793**	**350**	**44.4**	**37.5**	**51.3**		**785**	**319**	**40.4**	**33.6**	**47.3**	
**Age classes (years)**												
35–44	196	93	47.5	40.5	54.4	0.4272	191	77	40.3	33.4	47.3	0.6847
45–54	204	90	44.1	37.3	50.9		216	81	37.5	31.0	44.0	
55–64	219	99	45.2	38.6	51.8		214	91	42.5	35.9	49.2	
65–74	174	68	39.1	31.8	46.3		164	70	42.7	35.1	50.3	
**Italian Area**												
North	357	165	46.9	36.5	57.2	0.5635	359	144	39.5	29.5	49.6	0.8432
Centre	180	76	41.7	27.3	56.2		184	73	39.7	25.5	53.8	
South	256	109	42.8	30.7	54.8		242	102	42.3	30.0	54.7	
**Education**												
Primary and middle school	201	79	40.4	26.6	54.1	0.1252	209	84	39.6	26.2	53.0	0.9293
High school and university	591	271	45.8	37.8	53.8		575	235	40.8	32.8	48.8	
	**Cheese**											
	**N**	**n**	**%**	**CI 95%**		**χ^2^** ***p*-value**	**N**	**n**	**%**	**CI 95%**		**χ^2^** ***p*-value**
**All**	**793**	**355**	**44.6**	**37.7**	**51.5**		**785**	**421**	**53.7**	**46.8**	**60.6**	
**Age classes (years)**												
35–44	196	90	45.9	38.9	52.9	0.7559	191	96	50.3	43.2	57.4	0.7558
45–54	204	85	41.7	34.9	48.4		216	119	55.1	48.5	61.7	
55–64	219	102	46.6	40.0	53.2		214	116	54.2	47.5	60.9	
65–74	174	78	44.8	37.4	52.2		164	90	54.9	47.3	62.5	
**Italian Area**												
North	357	149	41.6	31.4	51.8	0.2765	359	189	52.8	42.5	63.1	0.6255
Centre	180	83	45.6	31.0	60.2		184	96	52.1	37.7	66.5	
South	256	123	48.2	36.0	60.4		242	136	56.3	43.9	68.6	
**Education**												
Primary and middle school	201	87	43.3	29.4	57.1	0.7006	209	123	58.8	45.4	72.3	0.0962
High school and university	591	267	44.9	37.0	52.9		575	298	52.0	43.9	60.1	

N: number of participants denominator. n: number of participants related to the prevalence numerator. For vegetables intake, the following foods were considered: leafy vegetables—cooked, leafy vegetables—raw, other vegetables, tomatoes—raw, tomatoes—cooked, root vegetables, cabbages, mushrooms, grain and pod vegetables, onion, garlic, stalk vegetables, mixed salad, mixed vegetables. For vegetables intake, the following foods were considered: leafy vegetable—cooked, leafy vegetable—raw, other vegetables, tomatoes—raw, tomatoes—cooked, root vegetables, cabbages, mushrooms, grain and pod vegetables, onion, garlic, stalk vegetables, mixed salad, mixed vegetables. For fruits intake, the following foods were considered: citrus fruits and other fruits. For fish intake, the following foods were considered: fish, crustaceans and molluscs. For cheese intake, the following foods were considered: cheeses (including fresh cheeses). Dietary information was collected by the self-administered Italian version of the food frequency questionnaire (FFQ) of the European Prospective Investigation into Cancer (EPIC). A balanced nutrition was defined based on the following intake: vegetables ≥ 200 g/day; fruit 150–375 g/day; fish at least twice per week (150 g per serving); cheese no more than three times per week (50–100 g per serving); sausages, salami and other preserved meat no more than once per week (50 g per serving); cake and desserts no more than once per week (100 g per serving); sugar beverages less than one per week (330 mL); and consumption of alcoholic beverages limited to two glasses per day for men (24 g of ethanol), one glass per day for women (12 g of ethanol) and one glass per day for men and women aged 65–74 (12 g of ethanol). SD: standard deviation; CI: confidence interval. Prevalence was age-standardized by the Italian National Institute of Statistics—ISTAT Italian population 2019 (except when it is reported by age-classes). Chi-square test to compare prevalence among classes. The pool was made of the following Italian regions: Piedmont, Lombardy, Liguria, Emilia Romagna, Tuscany, Lazio, Basilicata, Calabria and Sicily. Italian Area: North (Piedmont, Lombardy, Liguria, Emilia Romagna); Centre (Tuscany, Lazio); South (Basilicata, Calabria, Sicily).

**Table 4 healthcare-12-00475-t004:** Age-standardized (Italian population) balanced nutrition food groups intake (EPIC questionnaire): processed meat, sweets/cakes, sweet drinks and alcohol prevalence by sex, age classes, geographical area and educational level. Men and women residing in Italy aged 35–74 years, Health Examination Survey 2018–2019—CUORE Project.

	Men						Women					
	**Processed Meat**											
	**N**	**n**	**%**	**CI 95%**		**χ^2^ ** ** *p* ** **-value**	**N**	**n**	**%**	**CI 95%**		**χ^2^ ** ** *p* ** **-value**
**All**	**793**	**91**	**11.2**	**6.8**	**15.6**		**785**	**168**	**20.9**	**15.2**	**26.5**	
**Age classes (years)**												
35–44	196	12	6.1	2.8	9.5	0.0146	191	28	14.7	9.6	19.7	<0.0001
45–54	204	21	10.3	6.1	14.5		216	35	16.2	11.3	21.1	
55–64	219	30	13.7	9.1	18.3		214	52	24.3	18.6	30.1	
65–74	174	28	16.1	10.6	21.6		164	53	32.3	25.2	39.5	
**Italian Area**												
North	357	37	10.1	3.9	16.3	0.1479	359	83	22.4	13.8	31.0	0.4115
Centre	180	28	15.1	4.6	25.5		184	40	21.4	9.6	33.3	
South	256	26	10.0	2.7	17.4		242	45	18.2	8.5	27.8	
**Education**												
Primary and middle school	201	16	7.5	0.1	14.9	0.0913	209	54	24.5	12.7	36.2	0.0863
High school and university	591	75	12.4	7.1	17.7		575	114	19.6	13.2	26.1	
	**Sweets/Cakes**											
	**N**	**n**	**%**	**CI 95%**		**χ^2^ ** ** *p* ** **-value**	**N**	**n**	**%**	**CI 95%**		**χ^2^ ** ** *p* ** **-value**
**All**	**793**	**74**	**9.0**	**5.0**	**13.0**		**785**	**87**	**10.8**	**6.5**	**15.2**	
**Age classes (years)**												
35–44	196	11	5.6	2.4	8.8	0.0195	191	12	6.3	2.8	9.7	0.0051
45–54	204	14	6.9	3.4	10.3		216	18	8.3	4.7	12.0	
55–64	219	25	11.4	7.2	15.6		214	30	14.0	9.4	18.7	
65–74	174	24	13.8	8.7	18.9		164	27	16.5	10.8	22.1	
**Italian Area**												
North	357	29	7.9	2.3	13.5	0.1003	359	37	9.9	3.7	16.0	0.2929
Centre	180	13	7.0	0.0	14.4		184	17	8.7	0.6	16.8	
South	256	32	11.9	4.0	19.8		242	33	13.8	5.2	22.5	
**Education**												
Primary and middle school	201	14	6.8	0.0	13.8	0.2298	209	26	12.0	3.1	20.9	0.5529
High school and university	591	60	9.7	5.0	14.5		575	61	10.4	5.5	15.4	
	**Sweet Drinks**											
	**N**	**n**	**%**	**CI 95%**		**χ^2^ ** ** *p* ** **-value**	**N**	**n**	**%**	**CI 95%**		**χ^2^ ** ** *p* ** **-value**
**All**	**793**	**432**	**54.0**	**47.0**	**60.9**		**785**	**473**	**59.8**	**53.0**	**66.6**	
**Age classes (years)**												
35–44	196	81	41.3	34.4	48.2	<0.0001	191	87	45.6	38.5	52.6	<0.0001
45–54	204	108	52.9	46.1	59.8		216	123	56.9	50.3	63.6	
55–64	219	133	60.7	54.3	67.2		214	141	65.9	59.5	72.2	
65–74	174	110	63.2	56.1	70.4		164	122	74.4	67.7	81.1	
**Italian Area**												
North	357	194	53.7	43.4	64.1	0.8149	359	216	59.3	49.1	69.4	0.8341
Centre	180	95	52.8	38.2	67.4		184	108	58.2	44.0	72.5	
South	256	143	55.1	42.9	67.2		242	149	61.7	49.6	73.9	
**Education**												
Primary and middle school	201	106	51.3	37.3	65.3	0.6365	209	131	62.0	48.8	75.3	0.4669
High school and university	591	325	54.7	46.8	62.7		575	342	59.1	51.1	67.1	
	**Alcohol**											
	**N**	**n**	**%**	**CI 95%**		**χ^2^ ** ** *p* ** **-value**	**N**	**n**	**%**	**CI 95%**		**χ^2^ ** ** *p* ** **-value**
**All**	**793**	**581**	**74.2**	**68.2**	**80.3**		**785**	**662**	**84.3**	**79.3**	**89.4**	
**Age classes (years)**												
35–44	196	162	82.7	72.1	93.2	<0.0001	191	165	86.4	76.5	96.3	<0.0001
45–54	204	155	76.0	65.3	86.7		216	182	84.3	75.3	93.2	
55–64	219	167	76.3	64.9	87.6		214	178	83.2	73.2	93.2	
65–74	174	97	55.8	39.1	72.4		164	137	83.5	71.2	95.9	
**Italian Area**												
North	357	248	70.3	60.8	79.7	0.0438	359	292	81.5	73.5	89.5	0.0198
Centre	180	143	80.7	69.2	92.3		184	153	82.9	72.1	93.8	
South	256	190	75.2	64.6	85.8		242	217	89.5	81.8	97.1	
**Education**												
Primary and middle school	201	132	66.8	53.6	80.0	0.0058	209	182	86.6	77.3	95.9	0.2401
High school and university	591	449	76.8	70.1	83.6		575	479	83.5	77.5	89.5	

N: number of participants denominator. n: number of participants related to the prevalence numerator. For processed meat intake, the following foods were considered: sausages, salami and other preserved meat. For sweets/cakes intake, the following foods were considered: sugar, honey, jam, chocolate, candy bars, paste, confetti/flakes, non-chocolate confectionery, ice cream, cakes, pies, pastries, puddings (not milk-based), dry cakes and biscuits. For sweet beverages intake, the following foods were considered: fruit and vegetable juices, carbonated/soft/isotonic drinks and diluted syrups. For alcohol intake, the following foods were considered: alcoholic beverages. Dietary information was collected by the self-administered Italian version of the food frequency questionnaire (FFQ) of the European Prospective Investigation into Cancer (EPIC). A balanced nutrition was defined based on the following intake: vegetables ≥ 200 g/day; fruit 150–375 g/day; fish at least twice per week (150 g per serving); cheese no more than three times per week (50–100 g per serving); sausages, salami and other preserved meat no more than once per week (50 g per serving); cake and desserts no more than once per week (100 g per serving); sugar beverages less than one per week (330 mL); and consumption of alcoholic beverages limited to two glasses per day for men (24 g of ethanol), one glass per day for women (12 g of ethanol) and one glass per day for men and women aged 65–74 (12 g of ethanol). SD: standard deviation; CI: confidence interval. Prevalence was age standardized by Italian National Institute of Statistics—ISTAT Italian population 2019 (except when it is reported by age-classes). Chi-square test to compare prevalence among classes. The pool was made of the following Italian regions: Piedmont, Lombardy, Liguria, Emilia Romagna, Tuscany, Lazio, Basilicata, Calabria and Sicily. Italian Area: North (Piedmont, Lombardy, Liguria, Emilia Romagna); Centre (Tuscany, Lazio); South (Basilicata, Calabria, Sicily).

**Table 5 healthcare-12-00475-t005:** Age-standardized (Italian population) food group intake (EPIC questionnaire): vegetables, fruit, fish and cheese mean by sex, age classes, geographical area and educational level. Men and women residing in Italy aged 35–74 years, Health Examination Survey 2018–2019—CUORE Project.

	Men						Women					
	**Vegetable (g/day)**											
	**N**	**Mean**	**SD**	**CI 95%**		**ANOVA ** ** *p* ** **-value**	**N**	**Mean**	**SD**	**CI 95%**		**ANOVA ** ** *p* ** **-value**
**All**	**793**	**168.4**	**93.8**	**161.9**	**175.0**		**785**	**184.3**	**96.4**	**177.5**	**191.0**	
**Age classes (years)**												
35–44	196	163.1	84.2	151.3	174.9	0.0031	191	202.1	104.2	187.3	216.9	0.0031
45–54	204	179.4	106.0	164.9	194.0		216	188.3	90.5	176.2	200.4	
55–64	219	169.2	91.3	157.1	181.3		214	175.7	91.5	163.5	188.0	
65–74	174	155.3	86.0	142.6	168.1		164	166.6	99.9	151.3	181.9	
**Italian Area**												
North	357	167.2	91.7	157.7	176.7	0.0010	359	183.6	99.9	173.3	194.0	0.0010
Centre	180	182.9	89.8	169.8	196.1		184	200.1	98.2	185.9	214.2	
South	256	160.0	98.4	148.0	172.1		242	173.4	88.4	162.3	184.6	
**Education**												
Primary and middle school	201	156.6	79.8	145.6	167.6	<0.0001	209	162.7	85.4	151.1	174.3	<0.0001
High school and university	591	172.5	97.6	164.6	180.4		575	191.9	99.0	183.8	200.0	
	**Fruit (g/day)**											
	**N**	**Mean**	**SD**	**CI 95%**		**ANOVA ** ** *p* ** **-value**	**N**	**Mean**	**SD**	**CI 95%**		**ANOVA ** ** *p* ** **-value**
**All**	**793**	**268.1**	**174.7**	**255.9**	**280.2**		**785**	**275.7**	**165.2**	**264.1**	**287.2**	
**Age classes (years)**												
35–44	196	243.9	175.1	219.3	268.4	0.0013	191	270.6	145.2	250.0	291.2	0.0013
45–54	204	257.3	165.5	234.6	280.0		216	255.5	164.2	233.6	277.4	
55–64	219	290.1	183.2	265.8	314.4		214	294.2	166.7	271.9	316.5	
65–74	174	287.2	171.7	261.7	312.7		164	290.3	185.3	262.0	318.7	
**Italian Area**												
North	357	255.7	168.4	238.2	273.1	0.4007	359	276.5	163.1	259.6	293.4	0.4007
Centre	180	263.4	169.5	238.6	288.2		184	296.0	172.3	271.2	320.9	
South	256	288.6	185.4	265.9	311.3		242	259.2	161.5	238.9	279.6	
**Education**												
Primary and middle school	201	272.6	172.4	248.8	296.5	0.9245	209	267.4	156.8	246.1	288.6	0.9245
High school and university	591	266.7	175.7	252.5	280.9		575	278.4	168.1	264.6	292.1	
	**Vegetable and Fruit (g/day)**											
	**N**	**Mean**	**SD**	**CI 95%**		**ANOVA ** ** *p* ** **-value**	**N**	**Mean**	**SD**	**CI 95%**		**ANOVA ** ** *p* ** **-value**
**All**	**793**	**436.5**	**220.9**	**421.1**	**451.9**		**785**	**460.0**	**214.1**	**445.0**	**474.9**	
**Age classes (years)**												
35–44	196	407.0	219.1	376.3	437.7	0.3103	191	472.7	205.0	443.6	501.8	0.3103
45–54	204	436.7	209.3	408.0	465.4		216	443.8	208.7	416.0	471.6	
55–64	219	459.2	230.5	428.7	489.8		214	469.9	216.3	440.9	498.9	
65–74	174	442.5	225.6	409.0	476.1		164	457.0	231.8	421.5	492.4	
**Italian Area**												
North	357	422.9	218.3	400.2	445.5	0.0692	359	460.1	213.5	438.0	482.2	0.0692
Centre	180	446.3	210.0	415.6	477.0		184	496.1	225.7	463.5	528.7	
South	256	448.6	231.3	420.3	477.0		242	432.7	202.6	407.1	458.2	
**Education**												
Primary and middle school	201	429.2	208.5	400.4	458.0	0.0654	209	430.1	206.1	402.1	458.0	0.0654
High school and university	591	439.2	225.0	421.1	457.3		575	470.2	216.1	452.6	487.9	
	**Fish (g/day)**											
	**N**	**Mean**	**SD**	**CI 95%**		**ANOVA ** ** *p* ** **-value**	**N**	**Mean**	**SD**	**CI 95%**		**ANOVA ** ** *p* ** **-value**
**All**	**793**	**45.5**	**34.7**	**43.1**	**47.9**		**785**	**42.7**	**30.7**	**40.5**	**44.8**	
**Age classes (years)**												
35–44	196	46.6	35.1	41.7	51.6	0.6242	191	42.2	25.7	38.5	45.8	0.6242
45–54	204	45.3	33.7	40.6	49.9		216	42.1	33.0	37.7	46.5	
55–64	219	47.0	36.0	42.3	51.8		214	43.8	31.6	39.5	48.0	
65–74	174	42.0	33.9	37.0	47.0		164	42.8	31.3	38.0	47.6	
**Italian Area**												
North	357	47.7	36.5	44.0	51.5	0.3656	359	42.7	29.3	39.7	45.7	0.3656
Centre	180	41.2	28.9	37.0	45.5		184	43.1	32.9	38.4	47.9	
South	256	45.4	35.7	41.0	49.8		242	42.3	31.1	38.4	46.3	
**Education**												
Primary and middle school	201	42.5	33.9	37.8	47.1	0.0504	209	40.9	30.6	36.8	45.1	0.0504
High school and university	591	46.6	35.0	43.7	49.4		575	43.4	30.7	40.9	45.9	
	**Cheese (g/day)**											
	**N**	**Mean**	**SD**	**CI 95%**		**ANOVA ** ** *p* ** **-value**	**N**	**Mean**	**SD**	**CI 95%**		**ANOVA ** ** *p* ** **-value**
**All**	**793**	**42.7**	**34.2**	**40.4**	**45.1**		**785**	**37.0**	**31.2**	**34.8**	**39.2**	
**Age classes (years)**												
35–44	196	44.7	37.8	39.4	50.0	0.0997	191	41.2	36.2	36.1	46.4	0.0997
45–54	204	43.0	33.6	38.4	47.6		216	36.7	28.1	33.0	40.4	
55–64	219	42.4	33.1	38.0	46.8		214	35.6	29.5	31.7	39.6	
65–74	174	39.9	31.6	35.2	44.6		164	34.0	31.8	29.2	38.9	
**Italian Area**												
North	357	46.4	36.6	42.6	50.2	0.0169	359	38.4	33.2	35.0	41.9	0.0169
Centre	180	38.5	30.3	34.0	42.9		184	37.1	29.2	32.9	41.3	
South	256	40.6	32.9	36.6	44.7		242	34.8	29.7	31.1	38.6	
**Education**												
Primary and middle school	201	42.8	34.8	38.0	47.7	0.2342	209	33.8	32.8	29.3	38.2	0.2342
High school and university	591	42.7	34.0	40.0	45.5		575	38.1	30.6	35.6	40.6	

For vegetables intake, the following foods were considered: leafy vegetables—cooked, leafy vegetables—raw, other vegetables, tomatoes—raw, tomatoes—cooked, root vegetables, cabbages, mushrooms, grain and pod vegetables, onion, garlic, stalk vegetables, mixed salad, mixed vegetables. For vegetables intake, the following foods were considered: leafy vegetable—cooked, leafy vegetable—raw, other vegetables, tomatoes—raw, tomatoes—cooked, root vegetables, cabbages, mushrooms, grain and pod vegetables, onion, garlic, stalk vegetables, mixed salad, mixed vegetables. For fruits intake, the following foods were considered: citrus fruits and other fruits. For fish intake, the following foods were considered: fish, crustaceans and molluscs. For cheese intake, the following foods were considered: cheeses (including fresh cheeses). Dietary information was collected by the self-administered Italian version of the food frequency questionnaire (FFQ) of the European Prospective Investigation into Cancer (EPIC). SD: standard deviation; CI: confidence interval. Means and standard deviations were age standardized by Italian National Institute of Statistics—ISTAT Italian population 2019 (except when they are reported by age-classes). ANOVA to compare mean values among classes. The pool was made of the following Italian regions: Piedmont, Lombardy, Liguria, Emilia Romagna, Tuscany, Lazio, Basilicata, Calabria and Sicily. Italian Area: North (Piedmont, Lombardy, Liguria, Emilia Romagna); Centre (Tuscany, Lazio); South (Basilicata, Calabria, Sicily).

**Table 6 healthcare-12-00475-t006:** Age-standardized (Italian population) food group intake (EPIC questionnaire): processed meat, sweets/cakes, sweet drinks, and alcohol mean by sex, age classes, geographical area and educational level. Men and women residing in Italy aged 35–74 years, Health Examination Survey 2018–2019—CUORE Project.

	Men						Women					
	**Processed Meat (g/day)**											
	**N**	**Mean**	**SD**	**CI 95%**		**ANOVA ** ** *p* ** **-value**	**N**	**Mean**	**SD**	**CI 95%**		**ANOVA ** ** *p* ** **-value**
**All**	**793**	**36.4**	**28.8**	**34.4**	**38.4**		**785**	**24.4**	**21.6**	**22.9**	**25.9**	
**Age classes (years)**												
35–44	196	44.9	32.0	40.4	49.3	<0.0001	191	29.1	23.2	25.8	32.4	<0.0001
45–54	204	39.5	30.4	35.3	43.6		216	27.7	21.7	24.8	30.6	
55–64	219	32.8	25.9	29.4	36.3		214	21.2	21.4	18.3	24.1	
65–74	174	24.1	18.8	21.3	26.9		164	17.2	16.5	14.7	19.8	
**Italian Area**												
North	357	33.4	25.7	30.8	36.1	0.0011	359	23.7	21.4	21.5	25.9	0.0011
Centre	180	34.7	26.9	30.7	38.6		184	23.8	21.6	20.7	26.9	
South	256	41.6	33.1	37.5	45.7		242	25.9	22.0	23.1	28.6	
**Education**												
Primary and middle school	201	39.8	27.9	35.9	43.6	0.5038	209	23.3	22.1	20.3	26.3	0.5038
High school and university	591	35.2	29.0	32.9	37.6		575	24.7	21.3	23.0	26.5	
	**Sweets/Cakes (g/day)**											
	**N**	**Mean**	**SD**	**CI 95%**		**ANOVA ** ** *p* ** **-value**	**N**	**Mean**	**SD**	**CI 95%**		**ANOVA ** ** *p* ** **-value**
**All**	**793**	**84.1**	**59.8**	**80.0**	**88.3**		**785**	**77.5**	**61.8**	**73.2**	**81.8**	
**Age classes (years)**												
35–44	196	89.5	58.3	81.4	97.7	<0.0001	191	87.1	63.1	78.2	96.1	<0.0001
45–54	204	93.0	64.3	84.1	101.8		216	85.3	70.3	75.9	94.7	
55–64	219	79.9	57.1	72.4	87.5		214	70.8	54.9	63.4	78.1	
65–74	174	67.2	54.0	59.2	75.2		164	60.8	48.0	53.5	68.2	
**Italian Area**												
North	357	87.7	60.8	81.4	94.0	0.1249	359	80.5	59.3	74.3	86.6	0.1249
Centre	180	78.7	54.0	70.9	86.6		184	73.8	59.5	65.2	82.4	
South	256	82.8	62.1	75.2	90.4		242	75.9	67.0	67.4	84.3	
**Education**												
Primary and middle school	201	90.7	68.5	81.2	100.1	0.8558	209	73.7	59.4	65.6	81.7	0.8558
High school and university	591	81.7	56.2	77.1	86.2		575	78.7	62.7	73.6	83.8	
	**Sweet Drinks (ml/day)**											
	**N**	**Mean**	**SD**	**CI 95%**		**ANOVA ** ** *p* ** **-value**	**N**	**Mean**	**SD**	**CI 95%**		**ANOVA ** ** *p* ** **-value**
**All**	**793**	**99.8**	**159.7**	**88.7**	**111.0**		**785**	**72.6**	**130.0**	**63.5**	**81.6**	
**Age classes (years)**												
35–44	196	124.7	177.9	99.8	149.6	<0.0001	191	108.8	158.9	86.2	131.3	<0.0001
45–54	204	111.2	170.6	87.7	134.6		216	75.4	131.1	57.9	92.9	
55–64	219	81.8	133.7	64.1	99.5		214	59.4	121.0	43.2	75.6	
65–74	174	72.5	143.4	51.2	93.8		164	39.7	77.7	27.8	51.6	
**Italian Area**												
North	357	93.6	139.4	79.1	108.1	0.1505	359	73.8	129.1	60.4	87.1	0.1505
Centre	180	83.5	134.8	63.8	103.2		184	72.9	129.3	54.2	91.6	
South	256	120.0	196.6	95.9	144.0		242	70.6	132.3	53.9	87.2	
**Education**												
Primary and middle school	201	132.3	207.9	103.6	161.1	0.1004	209	64.5	114.2	49.1	80.0	0.1004
High school and university	591	89.4	139.1	78.2	100.6		575	75.2	135.2	64.2	86.3	
	**Alcohol (g/day)**											
	**N**	**Mean**	**SD**	**CI 95%**		**ANOVA ** ** *p* ** **-value**	**N**	**Mean**	**SD**	**CI 95%**		**ANOVA ** ** *p* ** **-value**
**All**	**793**	**14.7**	**16.4**	**13.5**	**15.8**		**785**	**5.5**	**9.1**	**4.9**	**6.1**	
**Age classes (years)**												
35–44	196	13.7	14.8	11.6	15.8	0.7067	191	6.1	10.4	4.6	7.6	0.7067
45–54	204	14.9	16.8	12.6	17.2		216	5.1	8.6	4.0	6.3	
55–64	219	14.4	17.0	12.2	16.7		214	5.4	8.5	4.3	6.6	
65–74	174	16.1	16.9	13.6	18.6		164	5.5	9.3	4.1	6.9	
**Italian Area**												
North	357	16.2	16.1	14.5	17.9	0.0037	359	6.7	10.4	5.6	7.8	0.0037
Centre	180	12.1	14.1	10.0	14.2		184	5.8	8.3	4.6	7.0	
South	256	14.4	17.9	12.2	16.6		242	3.5	7.2	2.6	4.4	
**Education**												
Primary and middle school	201	16.8	19.9	14.0	19.5	0.9708	209	4.2	8.1	3.1	5.2	0.9708
High school and university	591	14.0	15.0	12.8	15.2		575	6.0	9.4	5.2	6.8	

For processed meat intake, the following foods were considered: sausages, salami and other preserved meat. For sweets/cakes intake, the following foods were considered: sugar, honey, jam, chocolate, candy bars, paste, confetti/flakes, non-chocolate confectionery, ice cream, cakes, pies, pastries, puddings (not milk-based), dry cakes and biscuits. For sweet beverages intake, the following foods were considered: fruit and vegetable juices, carbonated/soft/isotonic drinks and diluted syrups. For alcohol intake, the following foods were considered: alcoholic beverages. Dietary information was collected by the self-administered Italian version of the food frequency questionnaire (FFQ) of the European Prospective Investigation into Cancer (EPIC). SD: standard deviation; CI: confidence interval. Means and standard deviations were age standardized by Italian National Institute of Statistics—ISTAT Italian population 2019 (except when they are reported by age-classes). ANOVA to compare mean values among classes. The pool was made of the following Italian regions: Piedmont, Lombardy, Liguria, Emilia Romagna, Tuscany, Lazio, Basilicata, Calabria and Sicily. Italian Area: North (Piedmont, Lombardy, Liguria, Emilia Romagna); Centre (Tuscany, Lazio); South (Basilicata, Calabria, Sicily).

**Table 7 healthcare-12-00475-t007:** Age-standardized (Italian population) food group intake (EPIC questionnaire): cereals, potatoes, legumes and oil mean by sex, age classes, geographical area and educational level. Men and women residing in Italy aged 35–74 years, Health Examination Survey 2018–2019—CUORE Project.

	Men						Women					
	**Cereals (g/day)**											
	**N**	**Mean**	**SD**	**CI 95%**		**ANOVA ** ** *p* ** **-value**	**N**	**Mean**	**SD**	**CI 95%**		**ANOVA ** ** *p* ** **-value**
**All**	**793**	**177.1**	**95.5**	**170.5**	**183.8**		**785**	**135.8**	**84.9**	**129.9**	**141.8**	
**Age classes (years)**												
35–44	196	176.6	96.9	163.0	190.2	0.1230	191	138.9	84.4	127.0	150.9	0.1230
45–54	204	169.8	89.0	157.5	182.0		216	138.9	89.1	127.0	150.8	
55–64	219	190.3	102.5	176.7	203.9		214	136.6	83.4	125.4	147.8	
65–74	174	170.2	92.1	156.6	183.9		164	124.9	79.7	112.7	137.1	
**Italian Area**												
North	357	181.5	99.8	171.1	191.8	0.5273	359	131.9	85.3	123.1	140.8	0.5273
Centre	180	176.6	88.2	163.7	189.5		184	144.9	82.2	133.0	156.8	
South	256	171.5	94.4	159.9	183.0		242	134.8	86.1	123.9	145.6	
**Education**												
Primary and middle school	201	178.5	91.6	165.8	191.1	0.0375	209	120.2	74.2	110.1	130.2	0.0375
High school and university	591	176.6	96.9	168.8	184.4		575	141.2	87.7	134.0	148.4	
	**Potatoes (g/day)**											
	**N**	**Mean**	**SD**	**CI 95%**		**ANOVA ** ** *p* ** **-value**	**N**	**Mean**	**SD**	**CI 95%**		**ANOVA ** ** *p* ** **-value**
**All**	**793**	**22.9**	**19.9**	**21.5**	**24.3**		**785**	**21.0**	**22.4**	**19.5**	**22.6**	
**Age classes (years)**												
35–44	196	25.8	19.1	23.1	28.4	<0.0001	191	25.0	29.5	20.8	29.2	<0.0001
45–54	204	24.9	19.5	22.2	27.6		216	24.0	21.4	21.2	26.9	
55–64	219	21.0	20.7	18.3	23.8		214	17.4	17.8	15.1	19.8	
65–74	174	18.2	19.4	15.4	21.1		164	15.9	17.3	13.3	18.6	
**Italian Area**												
North	357	22.4	20.2	20.3	24.5	0.0265	359	19.4	18.5	17.5	21.3	0.0265
Centre	180	23.4	20.4	20.4	26.4		184	19.6	20.0	16.7	22.5	
South	256	23.3	19.2	20.9	25.6		242	24.5	28.1	20.9	28.0	
**Education**												
Primary and middle school	201	21.4	19.7	18.6	24.1	0.1830	209	20.4	20.7	17.6	23.2	0.1830
High school and university	591	23.4	19.9	21.8	25.0		575	21.2	22.9	19.4	23.1	
	**Legumes (g/day)**											
	**N**	**Mean**	**SD**	**CI 95%**		**ANOVA ** ** *p* ** **-value**	**N**	**Mean**	**SD**	**CI 95%**		**ANOVA ** ** *p* ** **-value**
**All**	**793**	**21.6**	**18.9**	**20.3**	**22.9**		**785**	**19.7**	**15.9**	**18.5**	**20.8**	
**Age classes (years)**												
35–44	196	22.9	21.9	19.8	26.0	0.0710	191	21.0	17.9	18.4	23.5	0.0710
45–54	204	21.9	18.0	19.5	24.4		216	20.1	15.5	18.1	22.2	
55–64	219	21.3	18.8	18.8	23.8		214	19.4	15.4	17.3	21.5	
65–74	174	19.7	16.0	17.3	22.1		164	17.5	14.1	15.3	19.6	
**Italian Area**												
North	357	17.9	17.2	16.1	19.7	<0.0001	359	16.2	14.0	14.8	17.7	<0.0001
Centre	180	22.8	19.1	20.0	25.6		184	19.6	16.5	17.2	22.0	
South	256	26.0	20.1	23.5	28.4		242	24.7	16.7	22.6	26.8	
**Education**												
Primary and middle school	201	18.8	14.2	16.8	20.7	0.0069	209	18.3	14.9	16.3	20.4	0.0069
High school and university	591	22.6	20.2	21.0	24.2		575	20.1	16.2	18.8	21.4	
	**Oil (g/day)**											
	**N**	**Mean**	**SD**	**CI 95%**		**ANOVA ** ** *p* ** **-value**	**N**	**Mean**	**SD**	**CI 95%**		**ANOVA ** ** *p* ** **-value**
**All**	**793**	**15.9**	**11.4**	**15.1**	**16.7**		**785**	**17.4**	**11.2**	**16.6**	**18.2**	
**Age classes (years)**												
35–44	196	13.7	10.1	12.3	15.1	0.2151	191	19.0	11.8	17.3	20.6	0.2151
45–54	204	16.7	12.3	15.0	18.4		216	17.4	11.0	15.9	18.8	
55–64	219	17.6	11.4	16.1	19.1		214	16.7	10.9	15.2	18.2	
65–74	174	14.9	11.1	13.2	16.5		164	16.4	11.4	14.7	18.2	
**Italian Area**												
North	357	16.4	11.5	15.2	17.6	0.2149	359	17.4	11.6	16.2	18.6	0.2149
Centre	180	18.2	11.0	16.6	19.8		184	20.4	11.2	18.8	22.0	
South	256	13.5	11.2	12.2	14.9		242	15.1	10.3	13.8	16.4	
**Education**												
Primary and middle school	201	13.9	11.2	12.4	15.5	<0.0001	209	13.9	10.0	12.5	15.3	<0.0001
High school and university	591	16.5	11.4	15.6	17.5		575	18.6	11.4	17.7	19.6	

For cereals intake, the following foods were considered: pasta, rice, white and whole meal bread, other grains, crispbread, rusks and breakfast cereals. For potatoes intake, the following foods were considered: French fries, boiled potatoes, roasted potatoes, pure potatoes, croquette potatoes. Dietary information was collected by the self-administered Italian version of the food frequency questionnaire (FFQ) of the European Prospective Investigation into Cancer (EPIC). SD: standard deviation; CI: confidence interval. Means and standard deviations were age standardized by Italian National Institute of Statistics—ISTAT Italian population 2019 (except when they are reported by age-classes). ANOVA to compare mean values among classes. The pool was made of the following Italian regions: Piedmont, Lombardy, Liguria, Emilia Romagna, Tuscany, Lazio, Basilicata, Calabria and Sicily. Italian Area: North (Piedmont, Lombardy, Liguria, Emilia Romagna); Centre (Tuscany, Lazio); South (Basilicata, Calabria, Sicily).

**Table 8 healthcare-12-00475-t008:** Age-standardized (Italian population) food groups intake (EPIC questionnaire): meat, eggs, milk mean by sex, age classes, geographical area and educational level. Men and women residing in Italy aged 35–74 years, Health Examination Survey 2018–2019—CUORE Project.

	Men						Women					
	**Meat (g/day)**											
	**N**	**Mean**	**SD**	**CI 95%**		**ANOVA** ***p*-value**	**N**	**Mean**	**SD**	**CI 95%**		**ANOVA** ***p*-value**
**All**	**793**	**94.5**	**57.7**	**90.5**	**98.6**		**785**	**80.4**	**50.1**	**76.9**	**83.9**	
**Age classes (years)**												
35–44	196	104.2	57.7	96.2	112.3	<0.0001	191	88.7	53.7	81.1	96.3	<0.0001
45–54	204	100.4	57.9	92.4	108.3		216	87.7	51.2	80.9	94.6	
55–64	219	91.7	57.9	84.0	99.3		214	70.6	46.3	64.4	76.8	
65–74	174	74.7	52.5	66.9	82.5		164	71.5	45.0	64.6	78.4	
**Italian Area**												
North	357	93.9	57.5	88.0	99.9	0.7367	359	80.7	52.5	75.3	86.1	0.7367
Centre	180	94.4	55.4	86.3	102.4		184	83.8	48.8	76.7	90.8	
South	256	95.5	59.8	88.2	102.8		242	77.5	47.6	71.6	83.5	
**Education**												
Primary and middle school	201	99.5	58.5	91.4	107.6	0.6666	209	79.4	51.7	72.4	86.4	0.6666
High school and university	591	92.9	57.5	88.3	97.5		575	80.8	49.6	76.7	84.8	
	**Eggs (g/day)**											
	**N**	**Mean**	**SD**	**CI 95%**		**ANOVA** ***p*-value**	**N**	**Mean**	**SD**	**CI 95%**		**ANOVA** ***p*-value**
**All**	**793**	**10.9**	**10.1**	**10.2**	**11.6**		**785**	**10.1**	**8.5**	**9.5**	**10.7**	
**Age classes (years)**												
35–44	196	11.2	10.0	9.8	12.6	0.1783	191	10.6	9.1	9.3	11.9	0.1783
45–54	204	11.4	11.6	9.8	13.0		216	10.2	7.8	9.1	11.2	
55–64	219	10.9	9.5	9.6	12.2		214	9.9	9.3	8.7	11.2	
65–74	174	9.6	7.8	8.4	10.7		164	9.4	7.7	8.2	10.6	
**Italian Area**												
North	357	10.4	10.3	9.4	11.5	0.9016	359	10.7	9.6	9.7	11.7	0.9016
Centre	180	10.8	9.6	9.4	12.2		184	10.0	8.1	8.9	11.2	
South	256	11.6	10.1	10.4	12.8		242	9.2	6.9	8.3	10.1	
**Education**												
Primary and middle school	201	11.5	11.8	9.9	13.2	0.6666	209	9.4	8.2	8.3	10.5	0.6666
High school and university	591	10.7	9.4	9.9	11.5		575	10.3	8.6	9.6	11.0	
	**Milk (ml/day)**											
	**N**	**Mean**	**SD**	**CI 95%**		**ANOVA** ***p*-value**	**N**	**Mean**	**SD**	**CI 95%**		**ANOVA** ***p*-value**
**All**	**793**	**80.9**	**110.4**	**73.2**	**88.5**		**785**	**99.8**	**117.2**	**91.6**	**108.0**	
**Age classes (years)**												
35–44	196	71.3	99.3	57.4	85.2	0.6325	191	110.4	117.6	93.8	127.1	0.6325
45–54	204	73.0	105.5	58.5	87.5		216	108.0	126.0	91.2	124.8	
55–64	219	86.2	115.5	70.9	101.5		214	85.5	109.9	70.8	100.2	
65–74	174	100.2	123.6	81.8	118.5		164	93.0	109.5	76.3	109.8	
**Italian Area**												
North	357	81.8	110.4	70.4	93.2	0.4855	359	92.6	113.0	80.9	104.3	0.4855
Centre	180	80.3	102.5	65.4	95.3		184	108.4	117.0	91.5	125.3	
South	256	79.9	116.2	65.6	94.1		242	103.9	123.2	88.4	119.4	
**Education**												
Primary and middle school	201	78.8	105.6	64.2	93.4	0.9280	209	101.0	120.5	84.7	117.3	0.9280
High school and university	591	81.5	112.2	72.4	90.5		575	99.5	116.2	90.0	109.0	

For meats intake, the following foods were considered: beef, veal, pork, horse, chicken, turkey, rabbit (domestic). Dietary information was collected by the self-administered Italian version of the food frequency questionnaire (FFQ) of the European Prospective Investigation into Cancer (EPIC). SD: standard deviation; CI: confidence interval. Means and standard deviations were age standardized by Italian National Institute of Statistics—ISTAT Italian population 2019 (except when they are reported by age-classes). ANOVA to compare mean values among classes. The pool was made of the following Italian regions: Piedmont, Lombardy, Liguria, Emilia Romagna, Tuscany, Lazio, Basilicata, Calabria and Sicily. Italian Area: North (Piedmont, Lombardy, Liguria, Emilia Romagna); Centre (Tuscany, Lazio); South (Basilicata, Calabria, Sicily).

**Table 9 healthcare-12-00475-t009:** Age-standardized (Italian population) nutrients intake (EPIC questionnaire): proteins (total, animal and vegetable) mean by sex, age classes, geographical area and educational level. Men and women residing in Italy aged 35–74 years, Health Examination Survey 2018–2019—CUORE Project.

	Men						Women					
	**Proteins Tot (% Total Kcal)**											
	**N**	**Mean**	**SD**	**CI 95%**		**ANOVA ** ** *p* ** **-value**	**N**	**Mean**	**SD**	**CI 95%**		**ANOVA ** ** *p* ** **-value**
**All**	**793**	**15.5**	**3.1**	**15.3**	**15.7**		**785**	**16.0**	**3.6**	**15.8**	**16.3**	
**Age classes (years)**												
35–44	196	15.8	2.6	15.5	16.2	0.9283	191	15.6	2.7	15.2	16.0	0.9283
45–54	204	15.3	2.7	15.0	15.7		216	16.2	3.7	15.7	16.7	
55–64	219	15.5	3.0	15.1	15.9		214	15.9	3.8	15.4	16.4	
65–74	174	15.3	4.3	14.6	15.9		164	16.5	4.0	15.9	17.1	
**Italian Area**												
North	357	15.3	2.6	15.0	15.6	0.0975	359	15.9	3.3	15.6	16.2	0.0975
Centre	180	15.5	2.5	15.1	15.8		184	16.0	3.9	15.4	16.6	
South	256	15.8	3.9	15.3	16.2		242	16.3	3.8	15.8	16.8	
**Education**												
Primary and middle school	201	15.3	2.8	14.9	15.7	0.8686	209	16.2	3.4	15.8	16.7	0.8686
High school and university	591	15.6	3.2	15.3	15.8		575	16.0	3.7	15.7	16.3	
	**Animal proteins (% total Kcal)**											
	**N**	**Mean**	**SD**	**CI 95%**		**ANOVA ** ** *p* ** **-value**	**N**	**Mean**	**SD**	**CI 95%**		**ANOVA ** ** *p* ** **-value**
**All**	**793**	**10.4**	**3.6**	**10.1**	**10.6**		**785**	**10.8**	**4.1**	**10.5**	**11.1**	
**Age classes (years)**												
35–44	196	10.9	3.0	10.4	11.3	0.6278	191	10.6	3.1	10.1	11.0	0.6278
45–54	204	10.3	3.3	9.9	10.8		216	11.0	4.3	10.5	11.6	
55–64	219	10.2	3.5	9.7	10.7		214	10.6	4.4	10.1	11.2	
65–74	174	9.9	5.0	9.2	10.7		164	11.1	4.6	10.4	11.8	
**Italian Area**												
North	357	10.3	3.2	9.9	10.6	0.2704	359	10.8	3.8	10.4	11.2	0.2704
Centre	180	10.1	3.1	9.7	10.6		184	10.7	4.5	10.0	11.3	
South	256	10.7	4.5	10.1	11.2		242	11.0	4.2	10.4	11.5	
**Education**												
Primary and middle school	201	10.3	3.2	9.9	10.8	0.3210	209	11.2	3.8	10.7	11.7	0.3210
High school and university	591	10.4	3.8	10.1	10.7		575	10.7	4.2	10.4	11.1	
	**Vegetable proteins (% total Kcal)**											
	**N**	**Mean**	**SD**	**CI 95%**		**ANOVA ** ** *p* ** **-value**	**N**	**Mean**	**SD**	**CI 95%**		**ANOVA ** ** *p* ** **-value**
**All**	**793**	**5.1**	**1.2**	**5.1**	**5.2**		**785**	**5.2**	**1.2**	**5.1**	**5.3**	
**Age classes (years)**												
35–44	196	5.0	1.1	4.8	5.1	<0.0001	191	5.1	1.1	4.9	5.2	<0.0001
45–54	204	5.0	1.1	4.8	5.2		216	5.2	1.3	5.0	5.3	
55–64	219	5.3	1.2	5.2	5.5		214	5.3	1.2	5.1	5.5	
65–74	174	5.3	1.4	5.1	5.5		164	5.4	1.4	5.2	5.6	
**Italian Area**												
North	357	5.0	1.1	4.9	5.2	0.0011	359	5.1	1.2	5.0	5.2	0.0011
Centre	180	5.4	1.1	5.2	5.5		184	5.3	1.2	5.1	5.5	
South	256	5.1	1.3	4.9	5.3		242	5.3	1.3	5.2	5.5	
**Education**												
Primary and middle school	201	5.0	1.1	4.8	5.1	0.0060	209	5.0	1.3	4.9	5.2	0.0060
High school and university	591	5.2	1.2	5.1	5.3		575	5.3	1.2	5.2	5.4	

Dietary information was collected by the self-administered Italian version of the food frequency questionnaire (FFQ) of the European Prospective Investigation into Cancer (EPIC). SD: standard deviation; CI: confidence interval. Means and standard deviations were age standardized by Italian National Institute of Statistics—ISTAT Italian population 2019 (except when they are reported by age-classes). ANOVA to compare mean values among classes. The pool was made of the following Italian regions: Piedmont, Lombardy, Liguria, Emilia Romagna, Tuscany, Lazio, Basilicata, Calabria and Sicily. Italian Area: North (Piedmont, Lombardy, Liguria, Emilia Romagna); Centre (Tuscany, Lazio); South (Basilicata, Calabria, Sicily).

**Table 10 healthcare-12-00475-t010:** Age-standardized (Italian population) nutrients intake (EPIC questionnaire): lipids (animal and vegetable) and cholesterol mean by sex, age classes, geographical area and educational level. Men and women residing in Italy aged 35–74 years, Health Examination Survey 2018–2019—CUORE Project.

	Men						Women					
	**Animal Lipids (% Total Kcal)**											
	**N**	**Mean**	**SD**	**CI 95%**		**ANOVA ** ** *p* ** **-value**	**N**	**Mean**	**SD**	**CI 95%**		**ANOVA ** ** *p* ** **-value**
**All**	**793**	**17.9**	**5.3**	**17.6**	**18.3**		**785**	**17.6**	**5.2**	**17.2**	**17.9**	
**Age classes (years)**												
35–44	196	19.2	5.0	18.5	19.9	<0.0001	191	17.9	4.9	17.2	18.6	<0.0001
45–54	204	18.4	5.1	17.7	19.1		216	17.9	5.1	17.2	18.6	
55–64	219	17.2	5.2	16.5	17.9		214	17.3	5.4	16.6	18.0	
65–74	174	16.5	5.7	15.6	17.3		164	17.0	5.2	16.2	17.8	
**Italian Area**												
North	357	18.0	5.1	17.5	18.5	0.0074	359	17.7	4.9	17.2	18.2	0.0074
Centre	180	17.2	5.2	16.4	18.0		184	16.9	5.1	16.1	17.6	
South	256	18.4	5.6	17.7	19.1		242	17.9	5.6	17.2	18.6	
**Education**												
Primary and middle school	201	18.3	5.3	17.6	19.0	0.0096	209	18.5	4.9	17.8	19.2	0.0096
High school and university	591	17.8	5.3	17.4	18.2		575	17.3	5.2	16.8	17.7	
	**Vegetable Lipids (% total Kcal)**											
	**N**	**Mean**	**SD**	**CI 95%**		**ANOVA ** ** *p* ** **-value**	**N**	**Mean**	**SD**	**CI 95%**		**ANOVA ** ** *p* ** **-value**
**All**	**793**	**17.1**	**5.8**	**16.7**	**17.5**		**785**	**19.9**	**6.4**	**19.5**	**20.4**	
**Age classes (years)**												
35–44	196	16.5	5.1	15.8	17.2	0.1321	191	19.5	5.7	18.7	20.4	0.1321
45–54	204	17.7	6.1	16.9	18.6		216	20.2	6.7	19.3	21.1	
55–64	219	17.0	5.8	16.2	17.8		214	19.9	6.2	19.0	20.7	
65–74	174	17.3	6.0	16.4	18.2		164	20.1	6.9	19.1	21.2	
**Italian Area**												
North	357	17.2	5.9	16.6	17.8	0.0056	359	20.2	6.7	19.5	20.8	0.0056
Centre	180	18.1	6.0	17.3	19.0		184	20.2	6.5	19.3	21.1	
South	256	16.4	5.4	15.8	17.1		242	19.4	5.8	18.7	20.2	
**Education**												
Primary and middle school	201	15.9	5.7	15.1	16.7	<0.0001	209	18.8	5.8	18.0	19.6	<0.0001
High school and university	591	17.6	5.7	17.1	18.0		575	20.4	6.5	19.8	20.9	
	**Cholesterol (mg/day)**											
	**N**	**Mean**	**SD**	**CI 95%**		**ANOVA ** ** *p* ** **-value**	**N**	**Mean**	**SD**	**CI 95%**		**ANOVA ** ** *p* ** **-value**
**All**	**793**	**154.3**	**48.2**	**151.0**	**157.7**		**785**	**161.5**	**53.1**	**157.8**	**165.2**	
**Age classes (years)**												
35–44	196	160.7	44.2	154.5	166.8	0.2843	191	160.1	44.2	153.8	166.4	0.2843
45–54	204	156.9	43.2	150.9	162.8		216	162.7	50.4	156.0	169.4	
55–64	219	149.6	50.7	142.9	156.3		214	160.6	57.7	152.9	168.4	
65–74	174	148.2	56.2	139.9	156.6		164	162.5	61.0	153.1	171.8	
**Italian Area**												
North	357	150.6	43.0	146.1	155.0	0.0045	359	163.4	52.0	158.0	168.8	0.0045
Centre	180	151.2	47.6	144.2	158.1		184	153.3	49.7	146.1	160.5	
South	256	161.8	54.3	155.1	168.4		242	164.8	56.6	157.7	171.9	
**Education**												
Primary and middle school	201	157.5	43.1	151.6	163.5	0.0153	209	169.6	56.1	162.0	177.2	0.0153
High school and university	591	153.3	49.8	149.3	157.3		575	158.7	51.8	154.4	162.9	

Dietary information was collected by the self-administered Italian version of the food frequency questionnaire (FFQ) of the European Prospective Investigation into Cancer (EPIC). SD: standard deviation; CI: confidence interval. Means and standard deviations were age standardized by Italian National Institute of Statistics—ISTAT Italian population 2019 (except when they are reported by age-classes). ANOVA to compare mean values among classes. The pool was made of the following Italian regions: Piedmont, Lombardy, Liguria, Emilia Romagna, Tuscany, Lazio, Basilicata, Calabria and Sicily. Italian Area: North (Piedmont, Lombardy, Liguria, Emilia Romagna); Centre (Tuscany, Lazio); South (Basilicata, Calabria, Sicily).

**Table 11 healthcare-12-00475-t011:** Age-standardized (Italian population) nutrients intake (EPIC questionnaire): saturated, polyunsaturated and monounsaturated fat, lipids total mean by sex, age classes, geographical area and educational level. Men and women residing in Italy aged 35–74 years, Health Examination Survey 2018–2019—CUORE Project.

	Men						Women					
	**Saturated Fat (% Total Kcal)**											
	**N**	**Mean**	**SD**	**CI 95%**		**ANOVA** ***p*-value**	**N**	**Mean**	**SD**	**CI 95%**		**ANOVA** ***p*-value**
**All**	**793**	**11.0**	**2.4**	**10.8**	**11.1**		**785**	**11.4**	**2.4**	**11.2**	**11.6**	
**Age classes (years)**												
35–44	196	11.3	2.3	11.0	11.6	<0.0001	191	11.6	2.3	11.2	11.9	<0.0001
45–54	204	11.3	2.3	10.9	11.6		216	11.5	2.3	11.2	11.8	
55–64	219	10.6	2.4	10.3	11.0		214	11.3	2.6	10.9	11.6	
65–74	174	10.4	2.7	10.0	10.8		164	11.1	2.5	10.7	11.5	
**Italian Area**												
North	357	11.2	2.4	10.9	11.4	0.0048	359	11.5	2.3	11.3	11.8	0.0048
Centre	180	10.6	2.5	10.3	11.0		184	11.0	2.3	10.7	11.3	
South	256	10.9	2.5	10.6	11.2		242	11.5	2.7	11.1	11.8	
**Education**												
Primary and middle school	201	10.9	2.4	10.6	11.2	0.4720	209	11.6	2.4	11.3	11.9	0.4720
High school and university	591	11.0	2.5	10.8	11.2		575	11.3	2.5	11.1	11.5	
	**Polyunsaturated fat (% total Kcal)**											
	**N**	**Mean**	**SD**	**CI 95%**		**ANOVA** ***p*-value**	**N**	**Mean**	**SD**	**CI 95%**		**ANOVA** ***p*-value**
**All**	**793**	**5.0**	**1.3**	**4.9**	**5.1**		**785**	**5.3**	**1.4**	**5.2**	**5.4**	
**Age classes (years)**												
35–44	196	5.2	1.1	5.0	5.3	0.0069	191	5.3	1.2	5.1	5.4	0.0069
45–54	204	5.2	1.5	5.0	5.4		216	5.5	1.3	5.3	5.6	
55–64	219	4.8	1.3	4.7	5.0		214	5.3	1.3	5.1	5.4	
65–74	174	4.8	1.4	4.6	5.0		164	5.3	1.8	5.1	5.6	
**Italian Area**												
North	357	4.9	1.3	4.8	5.1	0.8683	359	5.4	1.5	5.3	5.6	0.8683
Centre	180	5.1	1.2	4.9	5.3		184	5.2	1.2	5.0	5.4	
South	256	5.1	1.5	4.9	5.2		242	5.3	1.3	5.1	5.5	
**Education**												
Primary and middle school	201	5.0	1.4	4.8	5.2	0.9168	209	5.4	1.3	5.2	5.5	0.9168
High school and university	591	5.0	1.3	4.9	5.1		575	5.3	1.4	5.2	5.4	
	**Monounsaturated fat (% total Kcal)**											
	**N**	**Mean**	**SD**	**CI 95%**		**ANOVA** ***p*-value**	**N**	**Mean**	**SD**	**CI 95%**		**ANOVA** ***p*-value**
**All**	**793**	**16.2**	**3.7**	**15.9**	**16.4**		**785**	**17.6**	**4.1**	**17.4**	**17.9**	
**Age classes (years)**												
35–44	196	16.2	3.1	15.8	16.6	0.0473	191	17.6	3.4	17.1	18.0	0.0473
45–54	204	16.7	3.8	16.1	17.2		216	17.9	4.3	17.3	18.5	
55–64	219	15.8	3.9	15.3	16.4		214	17.5	3.8	17.0	18.0	
65–74	174	15.7	3.8	15.1	16.3		164	17.5	4.7	16.8	18.2	
**Italian Area**												
North	357	16.1	3.6	15.7	16.5	0.2615	359	17.7	4.2	17.3	18.1	0.2615
Centre	180	16.6	3.7	16.0	17.1		184	17.7	3.8	17.1	18.3	
South	256	15.9	3.8	15.5	16.4		242	17.5	4.0	17.0	18.0	
**Education**												
Primary and middle school	201	15.4	3.6	14.9	15.9	0.0008	209	17.2	3.9	16.7	17.7	0.0008
High school and university	591	16.4	3.7	16.1	16.7		575	17.8	4.1	17.5	18.1	
	**Lipids Tot (% total Kcal)**											
	**N**	**Mean**	**SD**	**CI 95%**		**ANOVA** ***p*-value**	**N**	**Mean**	**SD**	**CI 95%**		**ANOVA** ***p*-value**
**All**	**793**	**35.1**	**6.6**	**34.6**	**35.5**		**785**	**37.5**	**6.6**	**37.1**	**38.0**	
**Age classes (years)**												
35–44	196	35.7	5.5	34.9	36.5	0.001	191	37.5	5.7	36.7	38.3	0.001
45–54	204	36.1	6.6	35.2	37.0		216	38.1	6.4	37.2	38.9	
55–64	219	34.2	6.9	33.3	35.1		214	37.2	6.4	36.3	38.1	
65–74	174	33.8	7.1	32.7	34.8		164	37.1	7.9	35.9	38.3	
**Italian Area**												
North	357	35.2	6.1	34.5	35.8	0.2901	359	37.9	6.6	37.2	38.6	0.2901
Centre	180	35.3	6.5	34.4	36.3		184	37.0	6.0	36.2	37.9	
South	256	34.8	7.2	33.9	35.7		242	37.3	6.9	36.5	38.2	
**Education**												
Primary and middle school	201	34.2	6.6	33.3	35.1	0.0593	209	37.3	6.4	36.4	38.2	0.0593
High school and university	591	35.4	6.6	34.9	35.9		575	37.6	6.6	37.1	38.1	

Dietary information was collected by the self-administered Italian version of the food frequency questionnaire (FFQ) of the European Prospective Investigation into Cancer (EPIC). SD: standard deviation; CI: confidence interval. Means and standard deviations were age standardized by Italian National Institute of Statistics—ISTAT Italian population 2019 (except when they are reported by age-classes). ANOVA to compare mean values among classes. The pool was made of the following Italian regions: Piedmont, Lombardy, Liguria, Emilia Romagna, Tuscany, Lazio, Basilicata, Calabria and Sicily. Italian Area: North (Piedmont, Lombardy, Liguria, Emilia Romagna); Centre (Tuscany, Lazio); South (Basilicata, Calabria, Sicily).

**Table 12 healthcare-12-00475-t012:** Age-standardized nutrients intake (EPIC questionnaire) fibre, sodium, potassium, carbohydrates and simple carbohydrates mean by sex, age classes, geographical area and educational level. Men and women residing in Italy aged 35–74 years, Health Examination Survey 2018–2019—CUORE Project.

	Men						Women					
	**Fibre (g/day)**											
	**N**	**Mean**	**SD**	**CI 95%**		**ANOVA** ***p*-value**	**N**	**Mean**	**SD**	**CI 95%**		**ANOVA** ***p*-value**
**All**	**793**	**9.9**	**2.7**	**9.7**	**10.0**		**785**	**11.3**	**3.0**	**11.1**	**11.5**	
**Age classes (years)**												
35–44	196	9.2	2.5	8.9	9.6	<0.0001	191	10.8	2.7	10.4	11.2	<0.0001
45–54	204	9.7	3.0	9.3	10.1		216	11.1	3.2	10.7	11.5	
55–64	219	10.2	2.5	9.9	10.5		214	11.7	3.0	11.3	12.1	
65–74	174	10.4	2.8	10.0	10.9		164	11.6	2.8	11.1	12.0	
**Italian Area**												
North	357	9.6	2.7	9.3	9.8	0.0049	359	11.1	3.1	10.8	11.4	0.0049
Centre	180	10.4	2.9	9.9	10.8		184	11.5	3.0	11.1	12.0	
South	256	9.9	2.7	9.6	10.2		242	11.3	2.8	11.0	11.7	
**Education**												
Primary and middle school	201	9.5	2.4	9.1	9.8	0.0358	209	11.0	3.1	10.6	11.4	0.0358
High school and university	591	10.0	2.8	9.8	10.2		575	11.4	2.9	11.1	11.6	
	**Sodium (g/day)**											
	**N**	**Mean**	**SD**	**CI 95%**		**ANOVA** ***p*-value**	**N**	**Mean**	**SD**	**CI 95%**		**ANOVA** ***p*-value**
**All**	**793**	**2.5**	**1.0**	**2.4**	**2.6**		**785**	**2.1**	**0.9**	**2.0**	**2.2**	
**Age classes (years)**												
35–44	196	2.7	1.1	2.5	2.8	<0.0001	191	2.3	1.0	2.2	2.4	<0.0001
45–54	204	2.6	1.0	2.5	2.7		216	2.2	0.9	2.1	2.3	
55–64	219	2.5	1.0	2.3	2.6		214	2.0	0.9	1.9	2.1	
65–74	174	2.2	0.9	2.0	2.3		164	1.8	0.9	1.6	1.9	
**Italian Area**												
North	357	2.5	1.0	2.4	2.6	0.9334	359	2.1	0.9	2.0	2.2	0.9334
Centre	180	2.4	0.9	2.3	2.6		184	2.2	1.0	2.0	2.3	
South	256	2.5	1.1	2.4	2.7		242	2.0	1.0	1.9	2.2	
**Education**												
Primary and middle school	201	2.6	1.1	2.5	2.8	0.2252	209	1.9	0.9	1.8	2.0	0.2252
High school and university	591	2.5	1.0	2.4	2.6		575	2.1	1.0	2.1	2.2	
	**Potassium (g/day)**											
	**N**	**Mean**	**SD**	**CI 95%**		**ANOVA** ***p*-value**	**N**	**Mean**	**SD**	**CI 95%**		**ANOVA** ***p*-value**
**All**	**793**	**3.2**	**1.0**	**3.1**	**3.3**		**785**	**3.0**	**1.0**	**3.0**	**3.1**	
**Age classes (years)**												
35–44	196	3.2	1.0	3.1	3.3	<0.0001	191	3.2	1.0	3.1	3.3	<0.0001
45–54	204	3.3	1.0	3.1	3.4		216	3.1	1.0	3.0	3.2	
55–64	219	3.2	1.1	3.1	3.3		214	2.9	1.0	2.8	3.1	
65–74	174	3.0	1.0	2.8	3.1		164	2.8	1.1	2.6	2.9	
**Italian Area**												
North	357	3.2	1.0	3.1	3.3	0.8018	359	3.0	1.0	2.9	3.1	0.8018
Centre	180	3.1	0.9	3.0	3.3		184	3.1	1.0	3.0	3.3	
South	256	3.2	1.2	3.1	3.4		242	2.9	1.1	2.8	3.1	
**Education**												
Primary and middle school	201	3.2	1.1	3.0	3.3	0.0008	209	2.8	1.0	2.6	2.9	0.0008
High school and university	591	3.2	1.0	3.1	3.3		575	3.1	1.0	3.0	3.2	
	**Carbohydrates (% total Kcal)**											
	**N**	**Mean**	**SD**	**CI 95%**		**ANOVA** ***p*-value**	**N**	**Mean**	**SD**	**CI 95%**		**ANOVA** ***p*-value**
**All**	**793**	**47.9**	**9.0**	**47.3**	**48.6**		**785**	**47.4**	**8.9**	**46.8**	**48.0**	
**Age classes (years)**												
35–44	196	47.3	7.8	46.2	48.4	0.2268	191	47.9	7.5	46.9	49.0	0.2268
45–54	204	47.3	8.9	46.0	48.5		216	46.9	8.6	45.7	48.0	
55–64	219	48.9	9.5	47.7	50.2		214	47.7	8.7	46.5	48.8	
65–74	174	48.5	9.9	47.0	50.0		164	47.4	11.1	45.7	49.1	
**Italian Area**												
North	357	47.5	8.4	46.6	48.3	0.0647	359	46.8	8.9	45.9	47.7	0.0647
Centre	180	48.4	8.2	47.2	49.6		184	47.8	8.1	46.6	48.9	
South	256	48.3	10.2	47.1	49.6		242	48.1	9.3	46.9	49.2	
**Education**												
Primary and middle school	201	48.6	9.1	47.3	49.8	0.1040	209	47.9	8.7	46.8	49.1	0.1040
High school and university	591	47.7	9.0	47.0	48.4		575	47.2	8.9	46.5	48.0	
	**Simple carbohydrates (% total Kcal)**											
	**N**	**Mean**	**SD**	**CI 95%**		**ANOVA** ***p*-value**	**N**	**Mean**	**SD**	**CI 95%**		**ANOVA** ***p*-value**
**All**	**793**	**18.6**	**6.2**	**18.1**	**19.0**		**785**	**20.1**	**6.7**	**19.7**	**20.6**	
**Age classes (years)**												
35–44	196	18.4	5.7	17.6	19.2	0.6066	191	20.9	6.4	20.0	21.8	0.6066
45–54	204	18.8	5.8	18.0	19.6		216	20.0	6.5	19.1	20.8	
55–64	219	18.4	6.4	17.5	19.2		214	20.2	6.3	19.3	21.0	
65–74	174	18.7	7.2	17.6	19.7		164	19.4	7.7	18.2	20.6	
**Italian Area**												
North	357	18.5	5.7	17.9	19.0	0.8063	359	20.5	6.8	19.8	21.2	0.8063
Centre	180	18.5	5.6	17.7	19.3		184	20.0	6.4	19.1	21.0	
South	256	18.8	7.2	17.9	19.7		242	19.7	6.7	18.8	20.5	
**Education**												
Primary and middle school	201	19.4	7.1	18.4	20.4	0.0033	209	20.9	7.0	20.0	21.9	0.0033
High school and university	591	18.3	5.9	17.8	18.8		575	19.8	6.5	19.3	20.4	

Dietary information was collected by the self-administered Italian version of the food frequency questionnaire (FFQ) of the European Prospective Investigation into Cancer (EPIC). SD: standard deviation; CI: confidence interval. Means and standard deviations were age standardized by Italian National Institute of Statistics—ISTAT Italian population 2019 (except when they are reported by age-classes). ANOVA to compare mean values among classes. The pool was made of the following Italian regions: Piedmont, Lombardy, Liguria, Emilia Romagna, Tuscany, Lazio, Basilicata, Calabria and Sicily. Italian Area: North (Piedmont, Lombardy, Liguria, Emilia Romagna); Centre (Tuscany, Lazio); South (Basilicata, Calabria, Sicily).

## Data Availability

The data are not publicly available due to ethical and legal restrictions on data sharing.

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
