# Peer review of "Nutrition, Physical Activity and Smoking Habit in the Italian General Adult Population: CUORE Project Health Examination Survey 2018–2019"

_healthcare, 2024, doi:10.3390/healthcare12040475_

Round 1

Reviewer 1 Report

Comments and Suggestions for Authors

The aim of the article titles:Nutrition, physical activity and smoking habit in the Italian general adult population: CUORE Project health examination survey 2018-2019” is to  evaluate lifestyles indicators, data collected in 2018-2019 in random samples of Italian adult general population, through the national HES implemented within the CUORE Project.

In general, the article is not well written, it is difficult to read, and the results section is very extensive and it is not clear. There are many repetitions throughout the text.

Abstract: please add the number of all participants, together.

Introduction:

This part is not comprehensive. The authors used only 4 references, and some of important ones, such as “WHO: State of Health in the EU” seems to be omitted. The authors could present the most important data from “OECD/European Observatory on Health Systems and Policies (2019), Italy: Country. Health Profile 2019, State of Health in the EU, OECD Publishing, Paris/European Observatory on Health Systems and Policies, Brussels” as an example of current health issues in the UE and in Italy, especially the results related to dietary risk, physical activity, tobacco use.

Line 51: please correct the format of the last sentence in the line 51 and next ones, when needed.

Results:

Is it necessary to repeat information about the project (in so detailed way) in the title of tables and figures?

Lines 203-206: is it correct to describe not significant results as “lower”. Please change similar phrases throughout the manuscript. There is a significant difference or there is no significant difference – it must be clear.

Please try to reduce the descriptions under tables and figures. Are all of them needed? For example, is duplicated description of the used method for defining a balanced diet needed under the Table 3? You mentioned that earlier in method section.

Is it justified to repeat the used methods few times as a result description?

In general, please focus on the most important results and describe them adequate. When there is a significant difference, please add the p-value. When there is no significant difference do not write that A is lower than B.

Discussion:

This part also needs to be improved. Please comment mentioned result in the same paragraph when your it is mentioned.

Lines: 570-573: when you describe a trend please add also the main results – what is the current fruit and vegetable consumption. Is it similar to your results?

Lines: 613-617: Was the blood pressure measured? Where are the results presented, since they are mentioned in conclusions?

Author Response

1. Abstract: please add the number of all participants, together.                    R.Thank you for the suggestion. The number of all participants was added.

Introduction:

2. This part is not comprehensive. The authors used only 4 references, and some of important ones, such as “WHO: State of Health in the EU” seems to be omitted. The authors could present the most important data from “OECD/European Observatory on Health Systems and Policies (2019), Italy: Country. Health Profile 2019, State of Health in the EU, OECD Publishing, Paris/European Observatory on Health Systems and Policies, Brussels” as an example of current health issues in the UE and in Italy, especially the results related to dietary risk, physical activity, tobacco use.                           

R.Thank you for the suggestion. Data from the “OECD/European Observatory on Health Systems and Policies (2019), Italy: Country. Health Profile 2019, State of Health in the EU, OECD Publishing, Paris/European Observatory on Health Systems and Policies, Brussels” were added in the introduction.

2. Line 51: please correct the format of the last sentence in the line 51 and next ones, when needed.

R.The word processor automatically formats justified text. I’m sorry, but it could not be changed.

Results:

4. Is it necessary to repeat information about the project (in so detailed way) in the title of tables and figures?

R. The tables and figures have been developed to be self-explanatory so as to make their extrapolation possible without necessarily having to refer to details contained in the text of the article. We have noticed that for the correct use of the results it is important to always give references to the name of the study, the period, the sex and age, and geographical area to which the data refers. This is particularly true, especially when results come from periodic studies conducted in many periods and similar tables and figures have already been published in reference to these previous studies.

5. Lines 203-206: is it correct to describe not significant results as “lower”. Please change similar phrases throughout the manuscript. There is a significant difference or there is no significant difference – it must be clear.

R. In the sentence and in the other similar sentences the word "lower" or "higher" are reported to describe in which direction the result is significant. In the case of the cited sentence, the difference between prevalences is statistically significant, as reported in the text, and the difference lies in the fact that elderly people have a lower prevalence compared to the prevalence in all class aof age considered together and in comparison to the first two age groups. The text has been changed to make it clearer.

6. Please try to reduce the descriptions under tables and figures. Are all of them needed? For example, is duplicated description of the used method for defining a balanced diet needed under the Table 3? You mentioned that earlier in method section.                                                                             

R.The tables and figures have been developed to be self-explanatory so as to make their extrapolation possible without necessarily having to refer to details contained in the text of the article. The article contains several age-and sex standardizations and many definitions so matching the data contained in the tables/figures to the methodology in the methods paragraph may be not simple and could lead to errors which may influence the interpretation of the results as well as their comparability with those from other studies.

7. Is it justified to repeat the used methods few times as a result description?

R. Thanks for noticing this. The tables/figures notes were incorrectly reported as the format of the results paragraph text. Formatting has been now changed.

8. In general, please focus on the most important results and describe them adequate. When there is a significant difference, please add the p-value. When there is no significant difference do not write that A is lower than B.

R. Thank you for the suggestion. The text has been modified in accordance with the suggestion to focus attention on the main results not reporting results that are not statistically significant. The processing of the results was set up by showing the confidence interval and using this to compare the statistics. Confidence intervals allow to evaluate the interval within which the value of the statistic falls at 95% in the population to which the statistical inference refers. If the confidence intervals of two same statistics calculated on different groups overlaps, the result is not significantly different, if it does not overlap it can be concluded that the result is statistically different. By showing confidence intervals it is therefore possible to make comparisons between statistics of different subgroups within the manuscript without the need to show p-values for all possible comparisons that may be of interest to the reader. Furthermore, through the confidence intervals it is possible for the reader to evaluate statistically significant differences also comparing results with similar data deriving from other studies.

Discussion:

9. This part also needs to be improved. Please comment mentioned result in the same paragraph when your it is mentioned.

R. Thank you for the suggestion. It is understandable that some sentences in the discussion may seem like a simple summary of the results, but, except for the necessary initial summary of the main results, the comments refer to the comparison of the results shown in this manuscript, with the results of the previous health examination surveys (1998-2002 and 2008-2012) and, when available, with the trends in Europe.

10. Lines: 570-573: when you describe a trend please add also the main results – what is the current fruit and vegetable consumption. Is it similar to your results?

R. Thank you for the suggestion. At the moment there are no other survey in Italy that estimate with national coverage the prevalence of fruit and vegetable consumption through the EPIC questionnaire (similarly for the other nutrients) and which therefore have comparable data. The data collected in this study - HES CUORE Project - are considered in the “Reference Intake Levels of Nutrients and Energy for the Italian population” (LARN). The temporal trends resulted by the comparison with the data collected in the previous CUORE Project Health Examination Survey (2008-2012) was reported, and the trend found in groups from other European countries was also reported.

11. Lines: 613-617: Was the blood pressure measured? Where are the results presented, since they are mentioned in conclusions?

R. Blood pressure was measured in all participants. The results were published (reference number 7): Donfrancesco C, Di Lonardo A, Lo Noce C, Buttari B, Profumo E, Vespasiano F, Vannucchi S, Galletti F, Onder G, Gulizia MM, Galeone D, Bellisario P, Palmieri L. Trends of blood pressure, raised blood pressure, hypertension and its control among Italian adults: CUORE Project cross-sectional health examination surveys 1998/2008/2018. BMJ Open. 2022 Nov 14;12(11):e064270.

Reviewer 2 Report

Comments and Suggestions for Authors

I congratulate the authors of the article for their great research. I understand that this article covers only a part of the total research project data. In it, the authors provide data on physical activity, nutrition and smoking. And the strength of the article is that it provides very detailed data on these research variables. In addition, the large study sample should be mentioned. So, overall, the article left a good impression. However, I would like to make a few comments, and perhaps the authors will take some of them into account.

Introduction

A brief introduction about non-communicable diseases and smoking, physical inactivity and incorrect eating, and it's completely understandable. Next, focus on an assessment of tobacco use, physical inactivity and dietary habits in the Italian adult general population through national Health Examination Surveys (HESs) implemented within the CUORE Project. Ok, but it remains unclear why this study should be relevant and interesting in a wider context, not only in Italy. Perhaps it is better to publish these data in a national scientific journal. Therefore, I recommend supplementing the introduction with a stronger justification of the relevance of this study in an international context.

Formulation of research aim is not very clear (to evaluate more recent <....> were analysed .....).

No research hypotheses?

Material and Methods

This section may be divided by subheadings. Information would be easier for readers to understand.

The HES 2018-2019 – CUORE Project used international standardized procedures and methods for measurements and data collection [5-8]. I looked at all these previous articles and it seems you used stratified random sample. But I missed more concrete information how each participant was invited to this study. Also, how informed consent was obtained. 

I found information about missing data screening.  However, I missed the information about the normality of the data. Therefore, it remains unclear why ANOVA was used to compare means (and why means were calculated).

Results

Maybe it could be good to include table (data) with information about the participants socio-demographics?

Title of the article “Nutrition, physical activity and smoking habit in the Italian ….”. Following such title, maybe it would be good to start from nutrition data, not smoking? But it just suggestion.

I think title of the figures have to be below figure.

I would consider if only p-values are enough?

Discussion

You are focusing on general prevalence of smoking, physical activity and nutrition, also comparisons by gender, educational level. But I missed some focus on age by commenting obtained results. 

Conclusions are clear. 

Author Response

I congratulate the authors of the article for their great research. I understand that this article covers only a part of the total research project data. In it, the authors provide data on physical activity, nutrition and smoking. And the strength of the article is that it provides very detailed data on these research variables. In addition, the large study sample should be mentioned. So, overall, the article left a good impression. However, I would like to make a few comments, and perhaps the authors will take some of them into account.

R. Thank you for understanding the enormous effort and desire to make essential health indicators available by basing the statistics on a large sample involved at a national level.

Introduction

1. A brief introduction about non-communicable diseases and smoking, physical inactivity and incorrect eating, and it's completely understandable. Next, focus on an assessment of tobacco use, physical inactivity and dietary habits in the Italian adult general population through national Health Examination Surveys (HESs) implemented within the CUORE Project. Ok, but it remains unclear why this study should be relevant and interesting in a wider context, not only in Italy. Perhaps it is better to publish these data in a national scientific journal. Therefore, I recommend supplementing the introduction with a stronger justification of the relevance of this study in an international context.

R. Thank you for the suggestion. In the introduction paragraph, the importance of providing updated indicators relating to risk factors and risk conditions of NCDs at an international level was added: “WHO recommended improving country-level surveillance and monitoring as a priori-ty in the fight against NCDs, also providing data disaggregated by age, gender, and socioeconomic groups [1, 2]. Monitoring should provide internationally comparable assessments of the trends in NCDs and related risk factors over time, help to benchmark the situation in individual countries versus others in the same region or development category, provide a support for advocacy, policy development and coordinated action [1, 2].”

2. Formulation of research aim is not very clear (to evaluate more recent <....> were analysed .....).No research hypotheses?

 R. Thank you for the suggestion. The aim of the survey is to provide internationally comparable assessments of lifestyles indicators. It has now been clarified in the sentence where the objectives are reported: “In order to provide more recent lifestyles indicators, data collected in 2018-2019 in random samples of Italian adult general population, through the national HES implemented within the CUORE Project, were analysed and reported by sex, age-classes, educational level, and geographical area.”

Material and Methods

3. This section may be divided by subheadings. Information would be easier for readers to understand.

 R. Thank you for the suggestion. The section has now been divided by subheadings.

4. The HES 2018-2019 – CUORE Project used international standardized procedures and methods for measurements and data collection [5-8]. I looked at all these previous articles and it seems you used stratified random sample. But I missed more concrete information how each participant was invited to this study. Also, how informed consent was obtained. 

 R. Thank you for the suggestion. This information has been added to the text: “[..] all participants received an invitation letter and an informative note by ordinary post-al service and signed an informed consent at the time of the visit.”

5. I found information about missing data screening.  However, I missed the information about the normality of the data. Therefore, it remains unclear why ANOVA was used to compare means (and why means were calculated).

 R. The data were collected from a large sample of the general population, therefore continuous variables distributions are bell-shaped, with only one peak, with kurtosis and skewness relating to the group or subgroup taken into consideration. The Central Limit Theorem assures that the distribution is approximately Normal so long as the sample size is large enough (about 30 units), allowing the use of much more powerful hypothesis tests and interval estimators than their non-parametric equivalents. In any case, use of means are not precluded. The indicators shown in the manuscript are usually presented as averages in national and international contexts and this ensures comparability between studies. For example, the average levels presented for nutrients are considered in the “Reference Intake Levels of Nutrients and Energy for the Italian population” (LARN).

Results

6. Maybe it could be good to include table (data) with information about the participants socio-demographics?

 R. Socio-demographics information include sex, age and social class. Educational level was used as a proxy for socioeconomic position. The tables show the number of participants in each stratum by gender, age and education classes. From these numbers it is therefore possible to evaluate the composition of the sample for these subgroups.

7. Title of the article “Nutrition, physical activity and smoking habit in the Italian ….”. Following such title, maybe it would be good to start from nutrition data, not smoking? But it just suggestion.

 R.Thank you for the suggestion. It is an understandable consideration, but we preferred to include nutrition in the title as the first lifestyle because it is actually the lifestyle for which the most data is presented, for the same reason we preferred to consider it in the text as the last lifestyle so as to don't suffocate other lifestyles with nutrition tables, which are definitely much more. We think that in this way it would be easier to consult the smoking and physical activity tables and at the same time make it clear from the title that nutrition is an important aspect in the manuscript.

8. I think title of the figures have to be below figure.

R. Thank you for the suggestion. The title of the figures has been placed below.

9. I would consider if only p-values are enough?

R. In all tables for each indicator the corresponding 95% confidence interval has been shown. By showing confidence intervals it is therefore possible to make comparisons between statistics of different subgroups within the manuscript without the need to show p-values for all possible comparisons that may be of interest to the reader. Furthermore, through the confidence intervals it is possible for the reader to evaluate statistically significant differences also comparing results with similar data deriving from other studies.

Discussion

10. You are focusing on general prevalence of smoking, physical activity and nutrition, also comparisons by gender, educational level. But I missed some focus on age by commenting obtained results. 

R. Thank you for the suggestion. A comment was added to the discussion on the lifestyle profile of the elderly which in fact is overall better than the general population: “Regarding the elderly, a better lifestyle profile was found compared to the general population, in fact the results show that they have tendencially lower prevalence of smoking habit and a higher prevalence of former smokers, significantly. lower prevalence of sedentariness during leisure time, significantly higher prevalence of healthy consumption of processed meat, sweet drinks, alcohol, significantly lower levels of total lipid, animal lipids, saturated fats, polyunsaturated fats and sodium and significantly higher levels of for fiber and vegetables proteins.”

Conclusions are clear. 

Reviewer 3 Report

Comments and Suggestions for Authors

In this manuscript, the authors present a comprehensive analysis of recent lifestyle indicators in Italy, focusing on modifiable risk factors for non-communicable diseases (NCDs) such as tobacco use, nutrition, and physical activity. The study utilizes data from the national Health Examination Survey conducted as part of the CUORE Project in 2018-2019. Key findings highlight a smoking prevalence of 23% for men and 19% for women, alongside gender disparities in sedentary behavior during leisure and work time. Furthermore, the investigation reveals notable differences in balanced eating behaviors, particularly regarding vegetable and fruit consumption, between men and women. Notably, the prevalence of a healthy lifestyle, defined as abstaining from smoking, engaging in regular physical activity, and adhering to at least five correct eating behaviors, remains low at 7% for men and 12% for women. This suggests persistently high levels of unhealthy lifestyles compared to a decade earlier, underscoring the imperative for sustained intersectoral strategies and ongoing monitoring efforts.

Additionally, a suggestion is proposed to replace the term "elderly" with "older adults" for enhanced inclusivity and precision in the manuscript's terminology.

Regarding the organization and presentation of the results, it is noted that the abundance of information may hinder readability. It is recommended to restructure the results section, aligning the sequence of tables and figures for improved coherence. Specifically, the alignment of supplementary tables related to physical activity and sedentary time, such as tables s2 and s15, could be reorganized for better accessibility and comprehension.

On line 75, it is suggested that the authors provide details about the physical activity questionnaire used and ideally include information regarding its validation for this specific population.

Furthermore, on line 153, a suggestion is made to consider incorporating a flowchart figure elucidating the participant selection process, enhancing the manuscript's clarity and reader comprehension.

Regarding discrepancies between textual descriptions and presented data in the tables, as noted on line 163, it is recommended that the authors ensure consistency and accuracy between the text and corresponding tables.

In addressing the comprehensive nature of the results section, a proposal is made to streamline the main text by retaining essential nutritional findings within the figures and relocating supportive data, such as tables, to the supplementary section. This could potentially enhance readability and focus.

Finally, in the discussion section, it is advised to consider restructuring the presentation by initiating with an introductory paragraph summarizing the most salient findings before delving into detailed comparisons across different parameters and years. This approach could facilitate a clearer understanding for readers, enhancing the coherence and flow of the discussion.

Comments on the Quality of English Language

Minor grammatical errors should be revised.

Author Response

1. Additionally, a suggestion is proposed to replace the term "elderly" with "older adults" for enhanced inclusivity and precision in the manuscript's terminology.

R. Thank you for the suggestion. The term elderly was replaced with older adults.

2. Regarding the organization and presentation of the results, it is noted that the abundance of information may hinder readability. It is recommended to restructure the results section, aligning the sequence of tables and figures for improved coherence. Specifically, the alignment of supplementary tables related to physical activity and sedentary time, such as tables s2 and s15, could be reorganized for better accessibility and comprehension.

 R. Thank you for the suggestion. The tables reported in the text contain standardized statistics with the age distribution of the adult Italian population. Supplementary tables include:

- some additional statistics compared to the tables reported in the text with the age distribution of the adult Italian population (from table S1 to table S13)

- the same tables reported in the text and in the supplementary materials but with standardization by age with standard European age distribution (from table S14 to table S25)

This dual standardization allows multiple uses and comparisons of statistics in varied contexts and for different purposes. The tables with the Italian standardization by age are those that are generally most used both in the national and international context as a reference for Italian data. For this reason, these tables were included in the text and as the first part of supplementary tables (from table S1 to table S13), and only subsequently reporting the supplementary tables relating to the European standard population (not real population used for some types of epidemiological comparisons) which are essentially a replication of the tables in the text and tables S1-S13 only with a different standardization by age.

The statistical analysis paragraph has been integrated in the following way in order to make the subdivision of the tables clearer:

“Following the suggestion reported in the WHO Global NCDs Action Plan 2013–2020 extended to 2030 [1, 2], indicators, where appropriate, were age standardized using the direct method, referring to the age- and sex-specific distributions of the Italian adult population in 2019 (Italian National Institute of Statistics-ISTAT) (Tables and figures, and supplemental materials tables S1-S13) [21]. Data were also age-standardized using the European Standard Population (EuStPop) 2013 for international comparisons (supplemental materials tables s14-s25) [22].”

3. On line 75, it is suggested that the authors provide details about the physical activity questionnaire used and ideally include information regarding its validation for this specific population.

 R. Thank you for the suggestion. Questions used in the physical activity questionnaire were added to the paragraph. As reported, the questionnaire was previously used in an Italian research project sponsored by the National Research Council and in the previous Italian HESs within the CUORE Project [5, 12].

4. Furthermore, on line 153, a suggestion is made to consider incorporating a flowchart figure elucidating the participant selection process, enhancing the manuscript's clarity and reader comprehension.

 R. Thank you for the suggestion. At the beginning of the results paragraph the description has been reformulated. In all tables, number of considered participants is reported for each strata. The number of participants used for the statistical analyses is reported for the three considered lifestyles. For physical activity and smoking habits, only participants for whom there was no information were excluded, for nutrition, participants without information on nutrition and those with the total energy consumption and the basal metabolic rate ratio outside the range considered appropriate were excluded.

5. Regarding discrepancies between textual descriptions and presented data in the tables, as noted on line 163, it is recommended that the authors ensure consistency and accuracy between the text and corresponding tables.

 R. Thank you for the suggestion. The sentence was changed and consistency and accuracy between the text and corresponding tables were checked choosing to delete from the text results that are not statistically significant.

6. In addressing the comprehensive nature of the results section, a proposal is made to streamline the main text by retaining essential nutritional findings within the figures and relocating supportive data, such as tables, to the supplementary section. This could potentially enhance readability and focus.

R.  Thank you for the suggestion. The nutrition-related indicators provided in the text tables are important reference points at national and international level. The data collected in this study - HES CUORE Project – for example are considered in the “Reference Intake Levels of Nutrients and Energy for the Italian population” (LARN). We prefer to include these tables in the text (sometimes supplementary materials are not considered by readers), and we chose this magazine also because it gives the possibility of including many tables and figures in the text, thinking that in the final formatting of the article, the graphics and layout will be better and will make it easier to read. In addition, the nutrition tables/figures notes in the text were incorrectly reported as the format of the results paragraph text, creating confusion. Formatting has been now changed.

7. Finally, in the discussion section, it is advised to consider restructuring the presentation by initiating with an introductory paragraph summarizing the most salient findings before delving into detailed comparisons across different parameters and years. This approach could facilitate a clearer understanding for readers, enhancing the coherence and flow of the discussion.

 R. Thank you for the suggestion. At the beginning of the discussion paragraph, the description relating to the three considered lifestyles was summarized, and, a summary description of the results relating to older adults who in fact appear to have a better profile than the rest of the general population was added.

Reviewer 4 Report

Comments and Suggestions for Authors

Dear authors,

First of all, thank you for the enormous effort that this type of study entails due to the large amount of sample that you have been able to access and all the information collected. It is very important to recognize the healthy lifestyle habits of the Italian population and observe if, indeed, greater access to education and information translates into a larger sector of the population having healthy habits. However, there are some comments that I would like you to take into account.

Summary:
- the abbreviation of the nutrition questionnaire appears in the methodology, and it has not appeared before.
- the statistics performed are not described.

Introduction:
- line 49: replace "," with and. (alcohol consumption and poor diet).
- what is the purpose of the study? What are the starting hypotheses?

Methodology:
- What is the questionnaire used to collect information on physical activity?
- From line 77 to 96, the text is a bit difficult to understand, could it be described using dashes?
- There is no specific section for statistics. And, generally, the statistical program is described in the first sentence.

Discussion:
- The first 3 paragraphs of the discussion are very descriptive. It is more like a results section, so some comparison with other studies should be added.

Tables and figures:
- Generally, the figure caption is written after the figure.
- The statistical procedure is described in the footer of the table, but it already appears in the table itself, so it would not be necessary.

Bibliography: in some references the month of publication appears, when it is not necessary. And, specifically, in number 27, "vol" appears.

Best regards.

Author Response

First of all, thank you for the enormous effort that this type of study entails due to the large amount of sample that you have been able to access and all the information collected. It is very important to recognize the healthy lifestyle habits of the Italian population and observe if, indeed, greater access to education and information translates into a larger sector of the population having healthy habits. However, there are some comments that I would like you to take into account.

R. Thank you for understanding the enormous effort and desire to make essential health indicators available by basing the statistics on a large sample involved at a national level.

1. Summary:
- the abbreviation of the nutrition questionnaire appears in the methodology, and it has not appeared before.
- the statistics performed are not described.

R. – Thank you for the suggestion. The questionnaire used is the European Prospective Investigation into Cancer (EPIC), a questionnaire internationally known among those who deal specifically with nutrition, so, having little space in the abstract, as for NCDs, we have directly reported the acronym. Since we cannot insert the full name for space reasons, we would still prefer to keep the acronym because it characterizes the questionnaire as very extensive and validated, and can help in the selection of manuscripts for meta-analysis or other types of comparisons.

- Thank you for the suggestion. In the methods section of the abstract it was added that the results are standardized by age and gender.

2. Introduction :
- line 49: replace "," with and. (alcohol consumption and poor diet).
- what is the purpose of the study? What are the starting hypotheses?

R. – Thank you for the suggestion. The sentence was now changed in “To intervene on the four main modifiable risk factors of NCDs (tobacco consumption, sedentary lifestyle/low physical activity, risky and alcohol consumption, and poor diet).), in Italy,….”

- The aim of the survey is to provide internationally comparable assessments of lifestyles indicators. It has now been clarified in the sentence where the objectives are reported at the end of the introduction paragraph: “In order to provide more recent lifestyles indicators, data collected in 2018-2019 in random samples of Italian adult general population, through the national HES implemented within the CUORE Project, were analysed and reported by sex, age-classes, educational level, and geographical area.”

In addition, in the introduction paragraph, the importance of providing updated indicators relating to risk factors and risk conditions of NCDs at an international level was added: “WHO recommended improving country-level surveillance and monitoring as a priori-ty in the fight against NCDs, also providing data disaggregated by age, gender, and socioeconomic groups [1, 2]. Monitoring should provide internationally comparable assessments of the trends in NCDs and related risk factors over time, help to benchmark the situation in individual countries versus others in the same region or development category, provide a support for advocacy, policy development and coordinated action [1, 2].”

3. Methodology:
- What is the questionnaire used to collect information on physical activity?
- From line 77 to 96, the text is a bit difficult to understand, could it be described using dashes?
- There is no specific section for statistics. And, generally, the statistical program is described in the first sentence.

R - Thank you for the suggestion. Questions used in the physical activity questionnaire were added to the paragraph. As reported, the questionnaire was previously used in an Italian research project sponsored by the National Research Council and in the previous Italian HESs within the CUORE Project [5, 12]. 

- Thank you for the suggestion.  Dashes were added to the text.                                    
- Thank you for the suggestion.  The materials and methods paragraph has now been divided into subparagraphs, including the statistical analysis paragraph.

4. Discussion:
- The first 3 paragraphs of the discussion are very descriptive. It is more like a results section, so some comparison with other studies should be added.

R. Thank you for the suggestion. Given the extent of the results, the usual summary of the main results at the beginning of the discussion paragraph is a bit long, but we believe it can be helpful in interpreting the results with a view to developing an overall picture of the Italian situation through the most recent data. In the following paragraphs of the discussion, the 2018-2019 results are compared with previous studies and trends are compared with those of other studies or international reports.

5.Tables and figures:
- Generally, the figure caption is written after the figure.
- The statistical procedure is described in the footer of the table, but it already appears in the table itself, so it would not be necessary.

R. - Thank you for the suggestion. The title of the figures has been placed below.

- In the notes of the tables and figures the abbreviations relating to the statistics (for example IC: confidence interval), some methods (age-sex standardization) and definition used are reported. ANOVA and chi-squared they were specified in the notes to underline that they are tests to overall compare values among class (age, geographical area and education) and not to compare men and women. The tables and figures with their titles and notes are designed to be self-explanatory.

6.Bibliography: in some references the month of publication appears, when it is not necessary. And, specifically, in number 27, "vol" appears.

R. - Thank you for the suggestion. Some corrections were done to the reference list.

Best regards.

Round 2

Reviewer 4 Report

Comments and Suggestions for Authors

Dear authors,
Thank you very much for attending the different considerations of the different reviewers. With the relevant modifications, work is better written, so it is better understood.
The considerations are lower, so only on line 21 there is a comma before a point and the font size of the tables is not the same in all.

All the best.

Author Response

Thank you for suggestions. Line 21 in now without comma. Regarding the format of tables we pasted and copied all from the same sources. I think that editorial office usually will reformat all tables according to the format of the journal.